# Quality of life in older immigrant adults on hemodialysis

Demba Keita *

Department of Interprofessional Health & Aging Studies, University of Indianapolis, Indianapolis, Indiana, United States of America

* keitad@uindy.edu, demba.aja84@gmail.com

## Abstract

### Background

In the United States (U.S.), over 34% of individuals with Chronic kidney disease (CKD) are aged 65 or older. Understanding the quality of life (QoL) in this population is essential. While there has been research on the experiences of U.S.-born older adults on hemodialysis, there is limited data on the experiences of older immigrant adults with CKD.

### Purpose

This study aims to explore the perceived QoL of older immigrant adults living with stage 5 CKD, with a focus on how the disease impacts their cultural beliefs, practices, and overall well-being.

### Methods

A qualitative, basic interpretive approach was employed to capture the lived experiences of older immigrant adults undergoing hemodialysis. Participants were selected based on predefined inclusion and exclusion criteria. The Short Blessed Test (SBT) was used to screen for cognitive impairment. Participants were recruited in the Mid-Hudson Valley Region of New York. Data were collected through semi-structured interviews. Thematic analysis was applied to the data to identify key themes and patterns in the participants' experiences.

### Results

The findings revealed that the QoL of older immigrant adults on hemodialysis is shaped by complex interactions between family and social support, cultural and religious practices, balancing independence and dependence, health and illness perceptions, life adjustments to hemodialysis, emotional responses, and immigration

**Data availability statement:** All relevant data are within the paper and its Supporting Information files.

**Funding:** The author(s) received no specific funding for this work.

**Competing interests:** The authors have declared that no competing interests exist.

adjustment. Participants highlighted the significance of spiritual and cultural beliefs in maintaining resilience and managing the emotional stress of the disease.

## Conclusion

The study underscores the need for culturally competent care that integrates social, emotional, and spiritual support to improve the QoL of older immigrant adults on hemodialysis. The findings suggest that CKD care must extend beyond biomedical factors to include sociocultural and spiritual dimensions. The results emphasize the importance of holistic healthcare approaches that respect cultural values and provide support systems to enhance the QoL of this vulnerable group.

## Introduction

CKD is a significant global health issue, especially among older adults, due to its progressive nature and its strong associations with comorbid conditions like cardio-vascular disease, diabetes, and hypertension [1,2]. Stage 5 CKD, also known as end-stage renal disease (ESRD), typically requires dialysis, which severely impacts physical, emotional, and social well-being [3]. While the burden of CKD is well-documented in the general population, the unique experiences of older immigrant adults with CKD in the U.S. have been underexplored.

Immigrant populations face distinct challenges, such as limited English proficiency, lower educational attainment, lack of health insurance, and cultural or religious bar-riers to care [4,5]. These challenges compound the physical and psychological toll of CKD, creating disparities in treatment engagement, patient autonomy, and overall health outcomes. Although previous research has examined the impact of hemodialy-sis on QoL, there is limited understanding of how cultural values, immigration status, and language differences influence QoL among older immigrant adults.

Globally, CKD affects approximately 13.4% of the population, with an especially high prevalence among older adults [6,7]. In the U.S., around 37 million people are affected, and CKD is a leading cause of cardiovascular morbidity and mortality [8]. The primary risk factors for CKD include aging, diabetes, obesity, and hypertension [1]. As kidney function deteriorates, patients may progress to ESRD, necessitating dialysis or a transplant.

QoL is a multi-dimensional concept that includes physical health, psychologi-cal well-being, independence, social relationships, and environmental context [9]. Hemodialysis significantly impacts daily life by imposing time-consuming treatments, fatigue, fluid restrictions, and dietary limitations [7,10]. For older adults, maintaining autonomy, spiritual well-being, and social engagement are often central to QoL [2].

Immigrant patients face specific challenges that can hinder their participation in shared decision-making and adherence to care plans. These challenges include language barriers, unfamiliarity with healthcare rights, and cultural norms that may conflict with biomedical practices [11]. For example, Muslim patients may choose to fast during Ramadan despite the health risks associated with CKD, highlighting the need for culturally sensitive care [12].

Undocumented immigrants, in particular, often lack access to early-stage CKD care, resulting in delayed treatment and more severe complications once they progress to ESRD [13]. These patients are more likely to rely on emergency care, which reduces QoL and increases healthcare costs. Additionally, many immigrants face isolation, stress, and discrimination, which further exacerbate their health challenges.

CKD is frequently accompanied by other chronic conditions, such as cardiovascular disease, diabetes, and protein-energy wasting (PEW), which worsen health outcomes [14]. Older adults with CKD are also at higher risk for cognitive impairment (CI), which may be exacerbated by dialysis-induced vascular stress and elevated uric acid levels [15,16]. CI can impair treatment comprehension and further complicate care.

Vascular access is a critical factor influencing health outcomes in dialysis patients. Arteriovenous (AV) fistulas, the preferred method of access, are associated with fewer complications and better survival rates compared to catheters [17,18]. However, many patients, especially older adults, initially receive catheters due to the immediate need for dialysis, despite the higher risk of complications [18].

Dialysis patients also commonly experience dermatological issues, such as uremic xerosis (dry skin), pruritus, and mucosal disorders, which significantly affect comfort and QoL [19,20]. Gastrointestinal symptoms, such as nausea, vomiting, and anorexia, further exacerbate malnutrition and complicate fluid management [21].

Psychological distress, including depression, is prevalent in dialysis patients and has been linked to poorer QoL, increased oxidative stress, and inflammation [22,23]. Hemodialysis treatment itself can worsen these symptoms due to the physical and emotional stressors associated with the treatment process [22].

Spirituality and cultural identity play central roles in coping with chronic illness for many immigrant patients. Religious practices such as prayer, fasting, and traditional healing methods provide emotional resilience but may also conflict with clinical guidelines, necessitating culturally sensitive healthcare practices [12,24]. Recognizing and respecting these beliefs is crucial for promoting QoL and improving health outcomes [12].

Older immigrant adults with stage 5 CKD represent a vulnerable yet under-researched population whose QoL is influenced by a combination of medical, cultural, linguistic, and socioeconomic factors. Future research and healthcare interventions must integrate cultural competence, social support, and individualized care planning to improve outcomes and reduce disparities for this growing demographic.

## Problem statement

The U.S. has a high prevalence of CKD, and 90% of people with CKD are unaware of their condition [25,26]. CKD among immigrants in the U.S. emerged as a significant public health concern, with studies indicating that immigrants faced a disproportionately higher risk of CKD than native-born populations [26,27]. This risk was particularly evident among older immigrants, a demographic that grew to approximately 7.1 million individuals, representing around 16% of the immigrant population in the U.S. [28,29]. Socioeconomic challenges, healthcare access limitations, cultural barriers, and health literacy disparities contributed to the vulnerability of this group, especially for immigrants from Latin American and Asian regions [29]. Despite this, research on CKD prevalence and its contributing factors among older immigrants remained sparse, leading to a substantial gap in knowledge and effective policymaking [30].

## Purpose statement

This study explored how older immigrant adults in the U.S. perceived the impact of living with stage 5 CKD on their QoL. Specifically, it aimed to identify and describe the social, economic, and cultural effects of CKD on this population, including how the disease influences their cultural beliefs and practices.

## Research questions

- How do older immigrant adults perceive the influence of living with stage 5 CKD on their QoL?

○ How do older immigrant adults describe the impact of living with stage 5 CKD on their cultural beliefs and practices?

## Definition of terms

- CKD: This disease is characterized by a gradual loss of kidney function, eventually leading to renal failure [31].

- Hemodialysis: A medical procedure used in patients with CKD (sometimes acute) to remove liquid waste and maintain electrolyte imbalance through a dialysis machine with the aid of a dialyzer [32].

- Older immigrant adults are 65 years old and older and have migrated to the U.S. as adults [33].

- QoL is the individual's perception of their position in life, including their goals, expectations, standards, and concerns relating to the culture and value systems within which they live [34].

- Stage 5 CKD: the most chronic stage of kidney disease, characterized by kidney failure, with an eGFR of less than 15, and dialysis is required to filter liquid waste and maintain electrolyte imbalance [35].

## Method

### Study design

A qualitative study using a basic interpretive design examined the QoL in older immigrant adults living with stage 5 CKD and how they described its effects on their cultural beliefs and practices. A basic qualitative interpretive approach was suitable for this study because it answered how people interpreted their experiences, constructed their worldviews, and what meaning they attributed to their experiences [36]. Qualitative research identifies trends and themes within text-based or observational data to describe and document the phenomenon of interest [37].

### Participants

This study's participants were older immigrant adults over 65 with stage 5 CKD who reside in the U.S.(Table 1). Before this study, participants must have received hemodialysis for at least three months (Table 3). Participants had to be able to read, write, and understand English. Patients who were born and raised in the U.S. were ineligible. Additionally, patients were excluded for cognitive impairment. Participation was entirely voluntary.

### Sampling and recruitment

The sample size for this study was guided by the concept of information power as proposed by Malterud et al. [38], which suggests that the more relevant information a sample holds in relation to the research aim, the fewer participants are needed. Information power is influenced by five key dimensions: (1) the aim of the study, (2) sample specificity, (3) use of theory, (4) quality of dialogue, and (5) analysis strategy.

The aim of this study was narrow and focused, exploring the lived experiences and quality of life of older immigrant adults living with stage 5 CKD in the United States—a population whose perspectives have been largely overlooked. The sample was highly specific, consisting of individuals meeting defined demographic and clinical inclusion criteria. Although the study was not based on a pre-existing theoretical framework, which would generally suggest the need for a larger sample, this limitation was offset by a strong focus in the study aim and consistency in interview delivery. Interviews were conducted using a semi-structured guide to ensure meaningful and comparable responses across participants.

Importantly, a significant number of participants provided rich, in-depth narratives, offering detailed insights into their personal, cultural, and clinical experiences. These comprehensive accounts contributed to a high level of information power. Furthermore, the primary researcher was supported by an experienced qualitative analysis supervisor who had

**Table 1. Participant Demographics (N = 11).**

| Category | Participants(n = 11) | Percent(%) |
|---|---|---|
| **Age** | | |
| 60-70 | 7 | 63.6 |
| Above 70 | 4 | 36.4 |
| **Gender** | | |
| Male | 7 | 63.6 |
| Female | 4 | 36.4 |
| **Ethnicity/Race** | | |
| White | 1 | 9.1 |
| Black | 4 | 36.4 |
| Hispanic | 1 | 9.1 |
| Asian | 2 | 18.2 |
| Arab | 3 | 27.3 |
| **Educational Level** | | |
| Less than High School/High School | 1 | 9.1 |
| College/University | 10 | 90.9 |
| **Country of Origin** | | |
| Bangladeshi | 2 | 18.2 |
| Jamaica | 1 | 9.1 |
| Nigeria | 1 | 9.1 |
| Tanzania | 1 | 9.1 |
| Ireland | 1 | 9.1 |
| Guayana | 1 | 9.1 |
| Egypt | 1 | 9.1 |
| Morocco | 1 | 9.1 |
| India | 1 | 9.1 |
| **Time In The USA (Years)** | | |
| ≤20 | 1 | 9.1 |
| 21–30 | 2 | 18.2 |
| 31–40 | 5 | 45.5 |
| 41–50 | 3 | 27.3 |

previously conducted studies using similar interpretive approaches, strengthening the reliability of the data interpretation. Taking all five dimensions into consideration, a target sample size of 10–12 participants was deemed appropriate to yield sufficient depth and variability of data to address the research aim.

Recruitment was initiated through in-person visits and phone calls to four dialysis centers in the Mid-Hudson Valley Region of New York, where older immigrant adults received hemodialysis. The primary researcher (D.K.) sent emails and, in some cases, hand-delivered formal request letters to the management of each dialysis center explaining the purpose and content of the proposed study, and permission to access patients was sought. Recruitment efforts included distributing one hundred flyers (Appendix C) across the dialysis centers. The flyers documented the study's aims and objectives and solicited participant involvement. The primary researcher's contact information was listed on the flyer, and individuals who wished to participate in the study could email or call if more details were needed. In addition, healthcare professionals throughout hemodialysis units were encouraged to distribute the flyer to older immigrant adult patients receiving hemodialysis treatment.

Even though there was no financial compensation for participating in this study, each participant received a $25 gift card as a symbol of gratitude for taking the time to participate. At the end of the interview, the gift cards were mailed to the participant's address using the U.S. Postal Service.

## Screening

To test for cognitive impairment among participants in this study, the primary researcher downloaded the free version of the Short Blessed Test (SBT) (Appendix B), which was available online. After potential participants consented to participate, the primary researcher screened each potential participant for cognitive impairment using the SBT via Zoom or phone call before collecting data. Potential participants responded to six questions on memory and concentration (Appendix B). SBT is sensitive to early cognitive changes associated with dementia (Han et al., 2022). A score of 0–4 indicated normal cognition, 5–9 indicated questionable impairment, and 10 or more indicates an impairment consistent with dementia [39]. Only participants with normal cognition were admitted to the study.

## Data collection

Data collection consisted of a semi-structured interview with each participant using an interview guide (Appendix A). Twelve open-ended questions explored participants' perceptions of living with stage 5 CKD, focusing on their QoL, cultural, spiritual, and religious beliefs. Interviews began as soon as the primary researcher received an email or a phone response from interested participants. Interviews were conducted via phone, Zoom, or Google Meet. All interviews were audio recorded using an iPhone 13 Pro Max application called Voice Recorder: Audio Editor software version 4.9. Zoom and Google Meet interviews were audio recorded using the Zoom or Google Meet audio recording application tools. The questions in the interview guide (Appendix A) explored the cultural and spiritual impacts of CKD and essential aspects of QoL.

The interview guide (Appendix A) included a section on demographics, an opening statement introducing the primary researcher, an explanation of the interview process, the purpose of the study, and the interview questions. The questions were designed to allow participants to elaborate as much as possible about their experiences with CKD and hemodialysis and its perceived impact on their QoL. The interview was audio recorded with the patient's consent to enhance accurate transcription and analysis. Each interview took approximately 45 minutes to one hour. After the interview, before data analysis, participants were contacted via phone and, or email to review transcripts to ensure their responses were accurately conveyed.

To protect participants' personally identifying information, numerical identifiers were used for each interview transcript, ensuring no personal data was linked to individual responses. Participants were not asked to provide sensitive information, including names, dates of birth, addresses, or social security numbers, following recommended guidelines for data privacy in qualitative research. All interview transcripts were digital and securely stored on the primary researcher's computer within a doubly password-protected file, accessible only to the primary researcher. No physical copies of transcripts were made to prevent unauthorized access. In compliance with data retention standards, the study transcripts are preserved on the researcher's computer for at least three years before secure deletion.

## Data management and analysis

At the end of the data collection phase, audio recordings of the interviews were converted into text using Temi software. The primary researcher de-identified and cleaned the transcripts by proofreading, listening, and comparing audio recordings with software-generated transcripts. All collected data were electronic and stored in the Dedoose software (Windows version 9.0.107-Installer) with an encrypted, password-protected file on the primary researcher's computer. Following data collection, data analysis took place.

This was done using the thematic analysis approach described by Kiger and Varpio (2020). The thematic analysis approach allows searching across a data set to identify, analyze, and report repeated patterns [40]. It allows data

description and interpretation in selecting codes and constructing themes [40]. Kiger and Varpio's (2020) guide to thematic analysis consisted of six steps, which are recursive rather than linear processes in which subsequent steps might prompt the researcher to return to earlier steps, considering new data or emerging themes that may warrant further investigation [40].

The first step in data analysis was to familiarize oneself with the data 40. This step in the thematic analysis process allowed the primary researcher to become familiar with the entire data set, which entailed repeated and active reading [40]. The data set in this study included interview transcripts, recorded observations, photographs, and videos.

The second step involved generating initial codes. Codes are an essential segment of the raw data that could be assessed meaningfully regarding the phenomenon [40]. In this step, the primary researcher, with the assistance of the analysis expert (L.S.), used the Dedoose software to generate initial codes. This step focused on identifying potential data items of interest, questions, connections between data items, and other preliminary ideas leading to a code book. In collaboration with the analysis expert, the primary researcher ensured that codes were sufficiently well-defined and demarcated, so they did not overlap with other codes and fit logically within a larger coding framework. In the third step, the primary researcher searched for themes using coded and collated data extracts. Themes were constructed by analyzing, combining, comparing, and graphically mapping how codes related, forming themes.

The fourth step involved reviewing themes to ensure adequate data support. This step included two phases. In the first analysis phase, the primary researcher looked at coded data within each theme to ensure proper fit. All relevant codes and data extracts under each theme were reviewed. To achieve this, Kiger and Varpio (2020) recommended that the primary researcher ensure the following questions were answered: Did each theme have adequate supporting data? Were the data included coherently to support the theme? Were some themes too large or diverse? Data within each theme had adequate commonality and coherence, and there was enough distinction between themes to merit separation.

At this point, data extracts were re-sorted, and themes were modified to reflect a better understanding of the participants' perspectives, which captured coded data to be added, combined, divided, or discarded. This first analysis phase was completed when the primary researcher was confident that the revised thematic map adequately covered all the coded data to be included in the final analysis. The primary researcher kept a detailed record of his decision-making process and thoughts about developing, modifying, and removing themes. As a result of these notes, the primary researcher created links between themes and an audit trail that reinforced the trustworthiness of the findings.

In phase two, similar questions were applied to the whole dataset. Here, the researcher determined whether individual themes were meaningful within the data set and whether the thematic map accurately reflected the whole data set. The primary researcher ensured that the thematic map demonstrated how themes interrelated and represented the primary research question. To achieve this task, the researcher reread the entire data set, coded for new data that fell under the themes developed or modified in this phase, and then revised the thematic map accordingly.

In the fifth step, the primary researcher defined and named themes. Once the thematic map had been refined, the primary researcher created a definition and narrative description of each theme, including why it was essential to the broader study question. As part of the primary researcher's review process, the names of themes for the final report were reviewed and ensured to be concise and adequately descriptive. The primary researcher investigated each theme to determine which aspects of the data set were covered by each theme. This way, a coherent narrative was created explaining how and why each coded theme contributed to understanding more significant questions and how it interacted with other themes. While addressing these questions, the researcher identified emerging sub-themes, identified areas of overlap between themes, and defined the scope of each theme and what it entails.

The sixth step concluded the analysis process by producing the final manuscript. This final step involved writing the final analysis and a description of the findings. Part of this process had been initiated in prior steps through notetaking, describing themes, and selecting representative data extracts. This step set the initial discussion stage and went

beyond codes and theme descriptions. Using narrative descriptions, representative data extracts, and direct quotations from participants, the final report provided a clear, concise, and logical account of how the primary researcher interpreted the data and why their interpretation was essential and accurate. As part of the analysis, the primary researcher described the data and explained how the explanation answered the research questions. Direct data extracts included adequate context to understand their meaning and were supported by a textual narrative from participants that explained their significance.

## Ethical statement

This study was conducted in accordance with ethical guidelines for research involving human participants. The research protocol, including the verbal consent procedure, was reviewed and approved by the University of Indianapolis Institutional Review Board (IRB), which granted an exemption determination under 45 CFR 46.104 (Exemption Date: January 8, 2024; Study Number: 01992). Informed verbal consent was obtained and documented via audio recording at the start of each interview. Participants were fully informed of the study's purpose, procedures, risks and benefits, confidentiality measures, and their rights, including the right to withdraw at any time without penalty. The primary researcher, who conducted all interviews, served as the witness to each consent. A standardized verbal script (Appendix D) was used to ensure consistency. The study posed minimal risk and followed the ethical principles of respect, beneficence, and justice outlined in the Belmont Report. All data were securely stored, and ethical oversight was maintained throughout the research process.

## Results

This study sample consisted of 11 older immigrant adults diagnosed with stage 5 CKD and undergoing maintenance hemodialysis. As shown in Table 1, participants ranged in age from 67 to 79, with the majority (63.6%) aged between 60 and 70 years. Males represented a larger proportion of the sample (63.6%) compared to females (36.4%).

The racial and ethnic composition of the group was notably diverse. Four participants (36.4%) identified as Black, three (27.3%) as Arab, two (18.2%) as Asian, one (9.1%) as White, and one (9.1%) as Hispanic. This diversity reflects the intersectionality of cultural identity and chronic illness among immigrant populations. Educationally, the majority of participants (90.9%) reported completing college or university, while only one individual (9.1%) had a high school level education or less, suggesting a relatively high level of formal education within this sample.

Participants originated from a wide range of countries, including Bangladesh (18.2%), Jamaica, Nigeria, Tanzania, Ireland, Guyana, Egypt, Morocco, and India (each comprising 9.1% of the sample). Time spent living in the U.S. varied, with the largest proportion (45.5%) having resided in the country for 31–40 years. Three participants (27.3%) had been in the U.S. for 41–50 years, two (18.2%) for 21–30 years, and one (9.1%) for 20 years or less.

This demographic profile illustrates the cultural and experiential heterogeneity of the sample, which provided a rich foundation for exploring the lived experiences of older immigrant adults with CKD. The combination of advanced age, lengthy U.S. residency, and diverse cultural backgrounds contextualizes the nuanced themes that emerged in the qualitative analysis—namely, family and social support, cultural and religious practices, balancing independence and dependence, health and illness perceptions, life adjustments to hemodialysis, emotional responses, and immigration adjustment (Table 2).

## Family and social support

Family and social support emerged as a central theme, with sub-themes encompassing emotional support, practical assistance, family bonds, and community involvement (Table 2). According to participant narratives, these elements significantly contributed to their psychological and physical well-being, helping them effectively manage the challenges of stage 5 CKD. Participants acknowledged the vital role of their families in alleviating the burdens associated with CKD.

Table 2. Themes, Sub-Themes, and Participant Quotes.

| Theme | Sub-Theme | Participant Quote |
|---|---|---|
| Family and Social Support | Emotional Support, Practical Assistance, Family Bonds, Community Involvement, | P-002: "On days when I feel too weak after dialysis, my niece steps in to help with laundry, groceries, and sometimes cooks for me. She usually does my groceries anyway. I don't even have to ask—she just knows."<br>P-006: "I live alone, but I'm never truly alone. My best friend of over 20 years is just a phone call away. When I need someone to run an errand or pick up my medication, I know I can count on him. I am thankful for him."<br>P-005: "My family has always been supportive. I feel this illness has enhanced my relationship with my kids; our bond is even stronger. My son calls every day, and my grandkids check in on weekends. That makes me feel really happy. I love my grandkids."<br>P-001: "We're blessed—my sister and two cousins live just five minutes away. Whether I need a ride to the clinic or just some company for dinner, they come in a heartbeat. That kind of closeness matters a lot to me."<br>P-007: "My daughter is truly exceptional. She rearranged her work schedule just to take me to dialysis. She cooks, keeps track of my meds, and even prays Maghreb with me. Her presence is my strength."<br>P-009: "My wife has been my steady hand through all this for the past 5 years. She manages my appointments, does our groceries, household chores, and everything, including signing all legal documents. Besides, she works part-time. Sometimes I feel it's too much for her."<br>P-008: "Since moving here, I've had no blood relatives around. But my friend Ali and his daughter have become my family. They visit, bring me food, and help around the house. Their kindness fills the gap. They are my family."<br>P-003: "Our community is tightly knit. When someone's in need, like when I had surgery, neighbors brought meals, offered rides, and even helped clean. We don't have to ask. People just show up." |
| Cultural and Religious Practices | • Spiritual Strength<br>• Cultural Heritage<br>• Religious Practices | P-002: "Yes, it does; I am a proud Bengali Muslim woman <laugh>, I consider myself a deeply religious, at least that is what my family says about me. My faith gives me the strength to get through dialysis. Wearing my veil is part of who I am, and nothing will change that."<br>P-010: "I am a strong Catholic, and that has been my support system in overcoming every challenge in life. It has not been easy, I can tell you that; however, going to church and talking to people really helps."<br>P-011: "I try to fulfill at least my basic religious obligations, as a Muslim, I have to pray five times a day, though I can't come to the Masjid for every single prayer; however, I try coming once or twice a week. Sometimes I feel down because I am not doing all those things I used to do."<br>P-005: "As a Chagga, I have been prepared for this. In my culture, there is a lot of discipline; we have been trained to ensure suffering at a very early age. Nothing breaks a Chagga, not even dialysis."<br>P-006: I have always been grateful to Allah for my health for the past 55 years. Before I had diabetes and later kidney disease, I was very healthy. Alhamdulillah (thank God), I still fast Ramadan and pray five times a day, and go for Jumah summons every Friday. So, I am thankful.<br>P-001: I love my Bengali dishes, Macher Jhol, Shorshe Ilish, Shukto, Dhokar, etc., umm, but most of these foods are unhealthy for me due to the high sodium and Potassium. I try to limit them, just eat them once in a while."<br>P-007: "I am an Arab Muslim, and religion and culture are just the same thing for me. I still eat my favorite Moroccan dishes. To be honest, nothing has changed as far as my eating habits."<br>P-008: "As a Muslim, Fridays are special days. After Jumah prayer, we gather in my house or my friends' for lunch and some-times dinner. Those things make me happy, so I look forward to Fridays. Those things remind me of the strength I have, and even though I'm far from home, I find peace in keeping those traditions alive." |

*(Continued)*

| Theme | Sub-Theme | Participant Quote |
|---|---|---|
| Balancing Independence and Dependence | • Maintaining Autonomy,<br>• Accepting Help | **P-011:** "I've always been an independent person. I do my lawn, clean, and even plow my driveway off snow. I've always prided myself on not needing help. But I've had to learn that asking for help isn't a sign of weakness. My illness has humbled me, and I've started to accept that it's okay to lean on others when I need to."<br>**P-001:** "There are times when I can manage everything on my own, but I've come to realize that relying on others sometimes is necessary, especially with my health. I used to do my laundry, clean around the house, but now all that is an ordeal, so I have to accept help from others. learned not to feel guilty about it; my wife and the entire Bengali community have been very supportive."<br>**P-002:** "I still try to run my errands, but when I'm not feeling well or if my energy is low, which is usually due to dialysis, my niece is always available to help. She will take time off from her work, just to make sure I get what I need.<br>**P-006:** "I've always been the one to handle everything myself, but now I'm finding that I need more help to get things done. Whether it's with household chores or managing my medical condition, I can't do it all alone anymore."<br>**P-003:** "Asking for help is something I've always struggled with. I would feel uncomfortable asking someone to do things for me, even when I know I can't manage it all. There's this feeling of burdening others, especially since I've been independent for so long. But I'm learning to let go of that pride, even if it's hard at times".<br>**P-004:** "Taking charge of my health has been a big step for me. I've started eating healthier, taking my medications regularly. Even though I'm managing my health better, I still lean on my family when it comes to other things, and that's okay."<br>**P-005:** "I've noticed that I need more help now than I used to. It's a difficult thing to admit because I've always tried to do everything on my own. But with the dialysis and the constant fatigue, I've had to accept that I can't keep up the same pace. My daughter drives me to my appointments. It's not easy, but I'm thankful for the support."<br>**P-007:** "I was able to manage everything by myself until dialysis became a regular part of my life. I had a routine that worked for me, but now things have changed. Dialysis takes up so much time, and the exhaustion afterward makes it hard to do the things I used to. I've had to ask for help with simple things, like cooking and laundry. It's humbling, but I've learned to accept it because I know I'm doing what's best for me." |
| Health and Illness | • Disease Onset<br>• Healthcare Services & Medical Coverage,<br>• Information Sharing,<br>• Diagnosis Experience,<br>• Physical Discomfort,<br>• Symptom Management, | **P-003:** "I'll never forget the day my kidney function test results came back. The doctor told me my kidney had failed, and it felt like the floor dropped from under me. I had always been healthy, and being a healthcare professional, I understood what CKD meant, but it was hard to accept it for myself. I tried to reconcile with the idea of dialysis, but it's been tough to get used to the routine. Every day is a struggle."<br>**P-005:** "By the way, before I was diagnosed with CKD, I had access to healthcare, which helped me manage my other health conditions. But after the diagnosis, it felt like everything changed. My health insurance didn't cover all the expenses, and I had to pay high premiums. It's been difficult to juggle appointments, treatments, and the financial strain that comes with it. I have to hire a maid to help with household chores."<br>**P-002:** "As a physician, I thought I knew everything about CKD. But when it hit me, it was a completely different experience. I knew what dialysis entailed, but I could never have imagined the physical and emotional toll it would take.<br>**P-009:** "I remember when my doctor called that blood work is back and that my eGFR is less than 15, so I had to start dialysis. At the time, I was not worried; however, with time, I hated dialysis. Dialysis is horrible, I'm hoping to get a transplant soon."<br>**P-006:** "At first, I didn't think much of the changes in my body. I noticed that I was getting tired more easily, and my urine looked darker than usual, but I didn't connect it to kidney disease. I thought it was just stress or aging, but then after a blood work, my provider called me, so he needed to discuss my blood work results. So, I went in, and he broke the news. Since then, everything has changed."<br>**P-007:** "It started gradually – my feet, face, and body began swelling, and I could feel it getting worse every day. Then, one day, I started having trouble breathing, which scared me. The swelling, the shortness of breath, it's hard to deal with. But I'm learning how to manage it, even though it's still overwhelming at times."<br>**P-009:** "The doctor explained that CKD could lead to a variety of complications, and I realized I needed to make changes in my diet. I love salt and I love sugar. I still struggle with my diet, but I can't help it."<br>**P-005:** "Before all this (CKD), I had high blood pressure and diabetes, which I thought I was managing well with medication until 2019 when I was diagnosed. Now, managing my kidney disease along with my other conditions feels like a constant battle, but I am grateful that I am alive." |

*(Continued)*

**Table 2.** (Continued)

| Theme | Sub-Theme | Participant Quote |
|---|---|---|
| Life Adjustments | • Lifestyle Changes,<br>• Activity Restrictions,<br>• Dietary Restrictions,<br>• Workforce Disruptions,<br>• Financial Hardships | **P-007:** "After my first dialysis, I met the social worker at the dialysis center. She recommended that I make changes to my diet, which included incorporating more vegetables, which, by the way, aligned with my cultural foods. For me, it wasn't just about following a medical diet; it was about reconnecting with my roots. I started incorporating more rice and lentils into my meals."<br><br>**P-005:** "Before CKD, I owned five restaurants, all of which were part of the community and an integral part of my identity. But once I was diagnosed, it became increasingly difficult to manage the stress of running the businesses. I had to make the tough decision to sell two of them, and one just went under because I couldn't keep up with the demands. It's been heartbreaking—food and hospitality were my life. Now, I'm just trying to adapt to this new reality where I have to rely on others to get by."<br><br>**P-002:** "Letting go of my favorite Bengali meals was one of the hardest things for me after my kidney diagnosis. I've always loved the rich flavors of my cultural dishes, biryani, puchkas, and sweets like rasgulla. So after my diagnosis, I had to cut down on a lot of the food I once enjoyed, especially the salt-laden dishes and fried snacks. It was a tough transition, as food has always been a big part of our family gatherings and celebrations. But now I try to find healthier alternatives that are still reminiscent of my childhood meals."<br><br>**P-008:** "Being constantly tired from dialysis has really taken a toll on my ability to keep up with my work. I used to be active, working in construction, but the fatigue has been overwhelming lately. Eventually, I had to make the difficult decision to resign from my job. It wasn't just about the physical exhaustion, but the mental burden of trying to stay focused while feeling drained. I never imagined that something like CKD could affect my work life so drastically, but now I've had to find other ways to occupy my time and manage my health."<br><br>**P-009:** "I've always worked hard to make a living. I used to live paycheck to paycheck, working long hours as a dishwasher in a restaurant. But after my CKD diagnosis, everything changed. The job I had was physically demanding, and my health made it impossible to continue. Not being able to work as I did before has been tough. It's not just the loss of income, but also the loss of my routine. The financial strain has been difficult, and I have to find ways to make ends meet without relying on physical labor."<br><br>**P-001:** "That is true because my wife and my daughter are highly restrictive for me. They stop me from having red meat. Even before kidney disease, we were very selective about meat, ensuring it had little or no fat. So yeah, it has restricted me from eating some foods, mostly anything with red meat, sweets, and starch."<br><br>**P-006:** "A lot has changed for me since I was diagnosed with CKD. I used to be someone who was always on the go, active, and involved with my community, but now I find myself limited by fatigue and physical discomfort. I can't do the things I used to do anymore—like walking for long distances or attending family events without feeling exhausted. My life has slowed down, and while it's frustrating, I'm trying to adjust to this new pace."<br><br>**P-011:** "My eating habits have changed with dialysis. I cannot drink as much fluid as I want to. I cannot eat our traditional Indian foods as much because they say you can't eat this; you can't eat that. So, some foods I can eat, some foods I can't, and I can eat a little bit of bananas. So, yes, it affected my choice of food." |

(Continued)

| Theme | Sub-Theme | Participant Quote |
|---|---|---|
| Emotional Responses | • Depression, • Frustration, • Faith, • Reflection | **P-008:** "The first few weeks were the hardest, and I was depressed, and even now, sometimes I struggle, I get emotional. But now I try to read more Quran, and that really helps, and I visit the masjid every evening, because I live very close by. "<br><br>**P-006:** "Dialysis is frustrating, and there are days when I feel like I'm not making any progress. I try to stay positive, but it's hard. Sometimes I just feel like I'm stuck in this cycle of uncertainty, and it can feel quite depressing."<br><br>**P-009:** "Going to dialysis each morning is tiring and difficult. I get frustrated sometimes; it's never-ending. But, amm, my faith in Allah has really helped me through all this. The five daily prayers really help."<br><br>**P-007:** "I was frustrated with dialysis, medications, and constant monitoring. But over time, I've come to terms with it. I told myself, 'This is my reality now,' and that helped me start adjusting to the changes. It's not easy, but I've found a way to make peace with it."<br><br>**P-011:** "As a physician, I knew exactly what my diagnosis meant, and that made it all the more difficult to accept. I understood the medical aspects, but emotionally, it was hard to face the reality. The frustration was overwhelming, especially since I had been in control of my own health for so long. Now, I had to rely on others, and that was a tough pill to swallow. I was depressed about what was happening to my body, and that feeling lingered for a long time. I've learned to cope, but it is still a struggle."<br><br>**P-005:** "At first, I was overwhelmed with a whirlwind of emotions—fear, frustration, and disbelief. But eventually, I had to accept it. 'This is my new life. I still struggle, but going to church, interacting with the community really helps."<br><br>**P-008:** "I have been depressed and frustrated. However, with the love and care from my family, I had to step up and be there for him, not just physically, but emotionally too. I may not have the same energy as I did before, but I'm determined to be there for him in whatever way I can."<br><br>**P-009:** "Some days are better than others, but there are times when I feel completely drained, physically and emotionally. I've learned to deal with it by taking things one day at a time. It's not always easy, and sometimes the frustration and sadness creep in, especially when I feel like I'm not progressing. But I try to focus on the positive, like spending time with my family or reading my favorite books. It's a tough journey, but I'm trying to keep moving forward, even if it's just one step at a time." |
| Immigration and Adjustment | • Professional Visa, • Citizenship Process | **P-002:** "For my husband and me, the immigration process wasn't as difficult as it might have been for others. We were both fortunate enough to qualify for a professional immigrant visa, which made the whole process smoother. I remember the relief we felt when we were approved; it gave us the stability we needed to start our lives here."<br><br>**P-010:** "It took some time, but eventually, I received my residency. At that point, I felt a great sense of accomplishment. That was enough for me. I didn't need to rush to citizenship—I was just grateful for the security of being a legal resident. My family was already here, and we could finally settle in."<br><br>**P-007:** "My journey to the U.S. citizenship was a bit different. My wife sponsored me because she was a U.S. citizen, which helped expedite the process. It was not an easy transition, but I always knew I had her support. At times, the process felt long and uncertain, but I trusted the system and believed it would eventually work out."<br><br>**P-006:** "The day I received my green card was an emotional moment for me. After years of waiting, I was finally granted permanent status. The best part was when I took the oath as a U.S. citizen—it made all the years of waiting and uncertainty feel meaningful. It changed my life."<br><br>**P-003:** "I came to the U.S. through a marriage visa—my wife had already become a U.S. citizen, and she sponsored me. The process was long, but it was made easier with her support. I'm proud of our journey, and today, I can say that being a U.S. citizen means much more than just a legal status; it's about belonging and the ability to tap opportunities as they arise."<br><br>**P-005:** "I came to the U.S. 30 years ago, as an asylee, and now that I am a U.S. citizen, I can truly say that my immigration status no longer affects my daily life. The years I spent waiting for approval, feeling unsure about the future, were challenging. But today, I feel like I am a part of this country."<br><br>**P-001:** "I came to the U.S. on a student visa, with dreams of completing my education and starting a new life. I didn't even have to try it. It was given to me because when I was doing my doctorate, one idea I was researching was that it was strategically important to the US government for technology. So I didn't even have to apply. My employer and the US government wanted to ensure that I didn't leave the U.S. after my doctoral program. So, I was given a green card, and finally, I took my oath as a U.S. citizen."<br><br>**P-004:** "My journey to U.S. citizenship was long and difficult, but through my wife's support, I eventually gained my green card. It was challenging, especially during the waiting periods, but I never gave up. When I finally became a U.S. citizen, I felt immense relief." |

## Emotional support

Most participants identified emotional support as a critical buffer against the psychological burden of stage 5 CKD and the demands of hemodialysis. Their experiences revealed a spectrum of emotional responses—ranging from resilience and gratitude to grief and profound isolation—shaped largely by the presence or absence of strong support networks.

Participant 004, a retired public health nurse, offered a heartfelt reflection on his daily struggle and the sustaining power of spousal support:

> "Without my wife's constant support and encouragement, I don't think I would have the strength to face this illness every day without her. When I'm down or sick after dialysis, she's right there—bringing me tea, helping me bathe, talking me through it. As you might know, dialysis is exhausting, so the little helps and care here and there make a huge difference."

Participant 003 emphasized a broader network of familial and social ties

> "Family comes first, no matter what. My daughter checks on me daily, and my close friends—they don't let me feel alone. Just knowing someone will pick up the phone at 2 a.m. if I need to talk, that's a lifeline."

Participant 002 spoke of the emotional shock following his diagnosis and how his family became his emotional anchor during that period:

> "I remember crying after I left the doctor's office—just sitting in the car alone, frozen. But when I told my family, they didn't let me spiral. My sister flew in the next week and stayed with me for a month. I felt cared for, and to be honest, that saved me."

Several participants also emphasized the importance of healthcare providers as sources of emotional stability. Participant 010 noted:

> "My relationship with my dialysis team is more than just medical—they've become my second family. They listen when I vent, and they celebrate when my numbers improve. That emotional connection makes all the difference."

Participant 003 described the isolating emotional toll of CKD, but also how community helped him cope:

> "Dealing with CKD has been emotionally draining. I often felt trapped in my body and couldn't see past the next session. But when I started talking to other patients in the waiting room—sharing frustrations, even laughing—it felt like I wasn't alone anymore."

Participant 004 added another layer, highlighting the interplay between faith and family:

> "There were days I felt like my world had turned upside down. The emotional rollercoaster was intense—fear, anger, sadness. But praying with my wife, hearing her say, 'We'll get through this,' gave me peace."

One of the most profound examples came from Participant 011, whose emotional journey was compounded by the traumatic loss of his wife to COVID-19 during the early phase of his dialysis treatment:

> "We both got COVID, and she didn't make it. She was supposed to be the one helping me through dialysis. I was devastated. I couldn't drive; I could barely function."

He went on to describe the pivotal role played by his extended family:

> "So, I had to depend on others to take me to dialysis. I couldn't drive at that time. So, my brother, sister, and nephew had to help me; my children lived a little far away. They also brought groceries, cooked food. That support… it kept me going."

Despite his loss, Participant 011 expressed a deep sense of gratitude

> "Emotionally, I felt supported—even in the darkest times. Without them, I don't know where I'd be."

### Practical assistance

Participants emphasized the importance of practical assistance from their families in managing the physical demands of treatment and daily activities associated with CKD. One participant emphasized his dependence on his wife and grandson, stating, "I rely heavily on my wife and grandson to help me with my medical appointments and errands" (Participant #007). According to Participant 011, the loss of his wife left him in a vulnerable position, requiring him to rely heavily on other family members for support. Unable to drive due to his condition, he had to depend on his brother, sister, and nephew for transportation to dialysis sessions and medical appointments. His children, though supportive, lived far away in other states, making it difficult for them to provide day-to-day assistance. Participant 011 recounted,

> My wife passed away from COVID-19. We both got COVID together. She died with it. So that was a big loss. She would have been my support system if I had renal failure. However, I am thankful for the support from my extended family and friends.

### Family bonds

Participant narratives echoed a solid family bond, which enhanced comprehensive support in managing their condition more effectively. Almost all participants emphasized the importance of strong family relationships, which provided emotional and practical support. Participant 001 shared, "My wife has been my rock through all of this; I don't know what I would do without her." Similarly, another participant highlighted how his diagnosis of CKD has strengthened his family ties, stating, "The diagnosis of CKD has strengthened my relationships with my family and loved ones" (Participant 001). Another participant reflected, "In our culture, familial bonds are paramount, and the collective support system plays a crucial role in navigating health challenges" (Participant 005). Additionally, Participant 003 spoke about the pivotal role of his daughter and a close friend, saying, "My daughter is my POA. My friend CO runs most of my errands and lives close by and he is available whenever I need help."

### Community involvement

Community involvement adds vital support layers, offering most participants a sense of belonging and purpose. Participant 005 highlighted how her participation in religious and cultural activities within her community enhanced her QoL, stating, "Being active in my community's religious and cultural events gives me strength and keeps me connected." Similarly, Participant 001 has a robust support network comprising close friends and relatives living nearby. This network provides a reliable source of assistance whenever needed. He mentions having relatives, such as cousins and brothers, and his daughter, who is always available to help with errands or other needs. This familial support system is not only readily accessible but deeply integrated into his daily life. Emphasizing the strong bonds within their community, he noted, "Our

family bonds are strong, but our social bonds are also very strong. If something happens, all of our close friends will jump in to help." Participant 003 also spoke about the support from his family and friends, mentioning,

> I have a supportive family I can rely on for assistance. If I needed someone to run an errand, I would likely call upon my daughter or son-in-law, who are usually available and willing to help.

### Cultural and religious practices

For most participants, cultural and religious practices are integral to their well-being, offering emotional comfort and a sense of community. Spiritual strength, cultural identity, and religious identity and practices were common sub-themes across participant narratives.

### Spiritual strength

Spiritual strength was crucial for most participants, providing comfort and resilience. A 70-year-old participant emphasized the importance of spiritual practices in maintaining mental well-being through personal prayers and reflections, Participant 007 stated,

> My prayers give me the strength to endure each day and remind me that I am not alone in this journey. And my faith teaches me that as long as we live, we will be healthy sometimes and be sick sometimes, it is all part of life.

Similarly, Participant 010 highlighted the sustaining power of faith, sharing, "My faith sustains me through the challenges of CKD and dialysis, reminding me that I am never alone, and that God's grace is sufficient for every trial and tribulation." Participant 004 echoed these sentiments, stating,

> I love to be spiritual all the time and my religion is an essential aspect of my life, it gives me comfort and strength during difficult times, especially with this kidney problem. To be candid with you, it has been tough over the past 2years, however, my faith and family support keep me going.

Five participants who practice the Islamic faith find solace and strength through personal prayers, which help them cope with the challenges of their illness. Similarly, four other participants who practiced Catholicism echoed similar experiences and maintained their faith through private spiritual activities, even though they could no longer participate fully in public religious events. One participant stated, "My faith has been a source of strength. Even when I cannot attend services, my prayers keep me going" (Participant 009). It underscores the importance of faith in providing emotional resilience and comfort for these participants. Spiritual belief has significantly influenced participants' perceptions of their illness and treatment. For many participants, faith provides a sense of community and emotional support, which has been vital for coping with the demands of hemodialysis. Participant 010: "My faith sustains me through the challenges of CKD and hemodialysis, reminding me that I am never alone and that God's grace is sufficient for every trial and tribulation." Participant 004: "My spirituality and religiosity are essential aspects of my life, providing comfort and strength during difficult times."

Two participants discussed how their participation in their local mosque activities helped them cope with CKD. One recounted, "On the two occasions that I was hospitalized, my niece went to the local mosque to have the Imam come to the hospital where I was hospitalized to pray for me. I consider myself a very religious and spiritual person; my spirituality is vital for my well-being" (Participant #002). Reflecting on his spirituality, Participant 001 stated,

> It's essential—very, very important. My blind faith in Allah keeps me going mentally. If you have faith in Allah and believe that whatever happens to you happens because of what Allah desires, then you have no worries. My faith in Allah is my fortress against depression. So many things become easy for you with that faith and belief.

he said, illustrating how religious beliefs provided strength and resilience.

### Cultural heritage

Participants expressed the profound role of cultural heritage in shaping their identity and providing a sense of belonging. Cultural heritage significantly influenced how most participants perceived and managed their illness. For example, Participant 004 relied on cultural values, including family support and traditional practices, stating,

> Our cultural traditions are very strong, and we care a lot about each other. My cultural foundation and the support from my family is what keeps me grounded and hopeful. In my culture, we support each other in difficult times and stand as a community to help whoever is in need.

Similarly, Participant 007 drew from his cultural background for emotional and practical support, proudly stating,

> I am an Igbo man, and nothing breaks an Igbo, not even dialysis. In my upbringing, I endured challenging childhood training away from my parents. These trainings break you and redesign you to face any challenges in life, so I am thankful to my Igbo traditions.

Additionally, Participant 003 emphasized the importance of cultural identity rooted in traditions, values, and beliefs passed down through generations. He explained, "My cultural identity is rooted in traditions, values, and beliefs passed down through generations. Spirituality is essential to me, and I find strength in my religious practices." Likewise, Participant 005 acknowledged the profound impact of cultural heritage on life, stating, "In my life, religious, spiritual, and cultural practices play a significant role in shaping my identity and providing a sense of belonging and meaning."

### Religious practices

Religious identity provided a profound sense of belonging and purpose for many participants. Most identified as Muslims or Catholics, and this foundation significantly shaped their worldview and coping strategies. Participant 009 shared, "My faith in Allah helps me through the hardest times, reminding me to stay strong and hopeful." Similarly, Participant 010 stated,

> As a woman of faith, I find solace in knowing that my worth and identity are not defined by my physical appearance but rather by my character, integrity, and relationship with God.

Participant 003 expressed how religious, spiritual, and cultural identities were deeply intertwined, stating,

> "I am a Muslim and that shapes my identity, both culturally and religiously, it plays a significant role in shaping who I am today. Every aspect of my life evolves around my faith as a Muslim."

Adapting religious practices to accommodate the demands of hemodialysis was crucial for maintaining continuity and purpose in the participants' lives. Despite the limitations imposed by their conditions, participants remained deeply committed to their faith through personal devotion, which helped sustain their spiritual well-being. One participant explained how hemodialysis affected participation in public religious events, such as Mass, stating, "I may not be able to attend Mass regularly, but I still pray every day and keep my faith strong" (Participant 003). Similarly, Participant #010 shared, "As a devout Catholic, my faith plays a central role in how I cope with the challenges posed by my health condition." Participant 004 reflected, "While the kidney problem may limit my ability to participate fully in certain religious practices, it does not restrict me from practicing my religion."

Participant 001 shared how CKD and hemodialysis affected his daily activities and religious practices, revealing a profound reliance on his faith. Despite his condition, he confidently stated, "It doesn't because I can still do everything for myself. My day-to-day activities are not affected by my condition." When asked about his religious, spiritual, and cultural identity, he expressed his deep commitment to Islam, emphasizing its central role in his life. Participant 001 explained.

> I am a Muslim; we take religion very strongly. We believe whatever happens for a purpose every single day in the morning. I always say one thing: I love you, my Lord, so guide me to what pleases you, and I need your forgiveness. Oh, my Lord, If you're happy with me, that's all that matters.

Religion remained a pillar in his life, providing guidance and comfort. He recounted the profound experiences of performing Hajj and Umrah with his family, Participant 001 stated,

> I was very fortunate to take my parents to Hajj 30 years ago and went with my wife 20 years ago. I have done Umrah 32 times. My children went to Umrah six times. So, when I think about it, I say it is not because of my position or financial ability but because of Allah's love and favor for me and my family.

This deep sense of gratitude underscored his unwavering faith. Reflecting on his health challenges, he expressed hope and faith in continuing religious practices. "Nowadays, I say I got stuck with this stuff: dialysis. I think I will go for Umrah or even Hajj when I get stable" (Participant 001), he said, showing his determination to maintain spiritual commitments despite his condition. When asked if CKD and dialysis impacted his religious practices, he affirmed, "No. No, it does not limit me in any way. I don't miss my prayers; I do everything religious obligation as required. My spiritual practice is my number one priority," he emphasized. However, he acknowledged the difficulty in fasting due to upcoming surgery and medications, highlighting practical adjustments without compromising his faith.

Participant 002's religious practices and identity profoundly impacted her life. Her faith provided comfort and resilience, and a vital aspect of her identity was proudly maintained despite societal challenges. She described the support network around her, particularly emphasizing assistance from her niece. While Participant 002 generally managed errands, her niece stepped in when necessary. This support was evident during hospitalizations when her niece sought solace by bringing the Imam from the local mosque to pray.

Reflecting on these experiences, Participant 002 stated,

> On the two occasions that I was hospitalized, my niece was scared and thought I was going to die. She went to the local mosque to have the Imam come to the hospital where I was hospitalized to pray for me.

Participant 002, expressed deep gratitude and pride for her niece, whom she adopted after her sister's death, showcasing their strong bond, she noted,

> I love my niece; I am so proud of her. She does not know her biological mother, my late sister. So, when my sister passed away due to cancer, my niece was just four years old. I adopted her, and we have been together for the past 26 years.

The presence of the Imam and the prayers were profoundly significant for her recovery. She explained,

> I consider myself a very religious and spiritual person, my spirituality is important to my well-being. Receiving the imam at the hospital energized me and made me very happy. I am very proud of my daughter.

This underscored how integral faith was to her health and resilience. Participant 002 also shared her experiences wearing a hijab in the U.S. She arrived when wearing a veil was uncommon and met with stares and comments. Despite initial feelings of discrimination, she grew accustomed to the remarks and remained steadfast in her faith, Participant 002 stated,

> I came to this country wearing a veil, which we call the 'Hijab,' at a time when this country was very conservative, and everyone called me the woman in the veil. In the beginning, it bothered me; however, as time went by, I got used to it, and it never bothered me. I have been wearing my veil my whole life; it is part of who I am, and nothing will change that, not even negative comments

### Balancing independence and dependence

Balancing independence and dependence is critical for most participants. While most participants desire autonomy in their day-to-day activities, they acknowledged the importance of accepting help.

**Maintaining autonomy.** Participants emphasized their strong desire to make autonomous decisions despite the challenges presented by their health conditions. For instance, Participant 001, a retired electrical engineer, shared,

> I drive myself to dialysis whenever I can, but when I'm too tired, I let my wife, or a ride-sharing service take over. It's important to me to keep as much of my independence as possible.

He also reflected on the decision-making process for his dialysis treatment, where his family's concerns influenced his choice of center-based dialysis over home treatment. "My nephrologist recommended home dialysis, but my family opposed it, so I had to respect my family's decision," he explained, adding, "My family cares so much about me."

This desire to balance independence with the need for assistance was echoed by Participant #009, who expressed, "I appreciate their assistance, but relying on others for my day-to-day needs can sometimes feel overwhelming." Similarly, Participant 002 noted, "Most of the time, I drive myself to my appointments. But if I can't, I am fortunate to have the support of family members and friends who assist me." Participant 011 added, "I do not look forward to dialysis; sometimes I wish I could just take a day off. Do I like dialysis? No, however, it is a necessary evil. " This sentiment reflected his wish for a more patient-centered approach that considers mental well-being alongside medical needs.

**Accepting help.** While difficult for most participants, accepting help became necessary for managing the complexities of their illness. Participants initially expressed reluctance to rely on others but eventually embraced assistance. One participant shared his experience of relying on his friend AK and AK's daughter for support. "I used to hate asking for help, but AK and my daughter have been a lifeline for me, especially with errands and doctor visits" (Participant 008). Accepting this help was instrumental in managing his condition. Similarly, Participant 004 described his routine and the need for occasional assistance, Participant 008 stated,

> I typically arrange transportation to my hemodialysis appointments, and I go three times a week. While I try to be independent, there are instances where I rely on family members or transportation services for assistance.

Another participant shared her experience of balancing independence and seeking help when necessary, Participant 002 shared,

> In terms of transportation to my appointments, most of the time, I drive myself. But if I can't, I am fortunate to have the support of family members and friends who assist me in getting to and from the dialysis center.

Participant 005 noted

> Yes, I do feel that I need more help now than before to get things done. The physical and emotional toll of managing CKD and undergoing hemodialysis treatments has made me reliant on assistance from family members, caregivers, and healthcare professionals.

Participant 010 added, "Depending on healthcare professionals and my family for support and assistance is an integral part of managing CKD." Another participant, a retired barber, experienced a sense of dependence on family for transportation and daily tasks, which sometimes felt overwhelming. "Relying on my family for rides and daily help was tough, but I try to keep things as normal as possible" (Participant 005). Despite the challenges, participants adapted to their routines and recognized the necessity of treatment for their health.

## Health and illness

For most participants, health and illness dynamics were central to their QoL. Disease onset, health services and medical coverage, diagnosis experience, physical discomfort, information sharing, and symptom management were common health and illness sub-themes among participants.

**Disease onset.** The onset of CKD was reported as a gradual and insidious process, with most participants typically unaware of their deteriorating kidney function until symptoms become pronounced. Four participants noted that prior to their diagnosis of CKD, their initial symptoms included significant leg swelling and high blood sugar levels. Participant 008 shared, "I was getting swollen, my friends noticed it, and also my wife". Participant 004, "When I first arrived in the U.S., my health was relatively good. A year before I was diagnosed with kidney disease, I experienced trouble breathing, and my blood sugar was high." Similarly, Participant 003 noted, "My daughter noticed that my face was swollen."

Another Participant 011, an internal medicine physician, recounts how a severe COVID-19 infection triggered his CKD. A month before his CKD diagnosis, he and his wife contracted COVID-19, resulting in a three-month stay in the ICU. Upon waking from a COVID-19-induced coma, he found himself on dialysis and was informed of his wife's death. Participant 011, shared,

> When I woke up from my COVID-19-induced coma, I was already hooked to a dialysis machine, and to make the situation even worse for me, when I woke up from the coma, I was also given the news of my wife's death.

According to Participant 011, before contracting COVID-19 and developing CKD, he had been managing diabetes for 20 years.

**Healthcare services and medical coverage.** While most participants find access to healthcare services easy, three participants highlighted the challenges of navigating the healthcare system and the importance of having access to regular medical care. All participants acknowledged the comprehensive explanations provided by their healthcare team, which helped them better understand their conditions. One participant stated, "When my nephrologist decided that I would start dialysis, three hours later, the dialysis center called and explained all the procedures and what hemodialysis entails" (Participant 001).

Participant 010, "The treatment procedures associated with hemodialysis, including the type of vascular access used for my treatments, have been thoroughly explained to me by my healthcare team." Participant 004, "Yes, the treatment procedures are explained to me by my healthcare providers, including details about vascular access and the dialysis process. Participants echoed a shared experience that access to healthcare services and reliable medical coverage was a guarantor for their overall QoL. All participants have at least one or more comorbidities prior to CKD diagnosis, requiring regular medical visits for maintenance care, which makes access to healthcare imperative for QoL

**Diagnosis experience.** The experience of receiving a CKD diagnosis is often distressing and overwhelming. All participants, except for one, described their diagnosis as a frightening and depressing event, initially feeling despair and hopelessness. One participant described, "When the doctor told me I had CKD, it felt like my world came crashing down. I was scared and didn't know what to expect" (Participant 005). This emotional response has been echoed among participants, as the diagnosis often comes with a significant lifestyle change and a need for ongoing medical treatment. Participant 004, "The news of my diagnosis was overwhelming and unsettling. It was difficult to come to terms with the fact that my kidneys were not functioning properly." Participant 010, "My overall experience with CKD and hemodialysis treatment procedures has been challenging yet manageable with the support of my healthcare team, family, and faith."

In recounting his diagnosis experience, one participant detailed a journey deeply intertwined with long-standing personal relationships with his healthcare providers. His primary care physician, a friend for 50 years, and his nephrologist, a friend for 40 years, have been closely monitoring his health. "All my healthcare providers are my close friends," he shared, emphasizing the trust and attentiveness in his care. His doctors, who have known him for decades, check on him every three months and quickly raise concerns if any health parameters shift. He became particularly vigilant about his creatinine levels and eGFR as indicators of his kidney function. He noted a troubling trend: "Suddenly, I saw that my creatinine was moving over 1.2, 1.4, 1.6. So, even 0.2 was significant from a percentage point of view" (Participant 001). When his creatinine spiked to 2.6, he realized his kidney function was declining rapidly. "When eGFR goes up, eGFR comes down reverse, and eGFR comes down from 32 to 15. Yeah, that is a dead kidney," (Participant 001) he explained, understanding the gravity of his situation.

The critical moment came three months ago during a trip to Manhattan for a work meeting with his daughter. His nephrologist, who rarely contacts him directly, called urgently: "Where are you now, exactly?"(Participant 001) Realizing the seriousness of the call, he immediately agreed to turn around and meet his doctor at the hospital. "I knew exactly what was wrong because I had seen the report the night before," Participant 001 recounted, having received his lab results from Quest Diagnostics ahead of his doctor.

Upon arrival at the hospital, his nephrologist confirmed his fears: his kidney function had plummeted from two to 7.5 in just two months. "I want to check you in," his doctor said, and soon he found himself admitted to the emergency room. Accompanied by his daughter, he underwent a series of procedures, including the insertion of a central catheter, a necessary step for the commencement of dialysis. By ten o'clock that night, he was transported to the dialysis center to begin treatment. Exactly three months ago, he started dialysis, he reflected, marking the beginning of a new chapter in managing his CKD. This participant expressed gratitude for the personal relationships with his healthcare team and explained the critical role of vigilant monitoring, timely intervention, and the profound impact of supportive healthcare relationships in navigating life-altering diagnoses.

**Physical discomfort.** Most participants have been on hemodialysis for over 5 years (Table 3). Participants who have been on hemodialysis for 4 years or more reported more symptoms that tend to affect their QoL. Participant 008, who has been on hemodialysis for 8 years, shared, "I have every issue that you can think of in a patient: my body hurts all over, I am mostly constipated and went through an open-heart surgery two years ago, I am alive by a miracle." Also, Participant# 009, who has been on hemodialysis for 5 years, noted: "There are several issues I encounter with hemodialysis, including fatigue, cramps, constipation, and low blood pressure during or after hemodialysis." Further, Participant 002, who has been on dialysis for 6 years, shared, "In addition to the logistical challenges of hemodialysis treatment, I encounter some issues like being fatigued, dizziness, and fluctuations in blood pressure." A common symptom experienced by all participants is the pain associated with the insertion of needles during dialysis and general physical exhaustion following each session. Participants described the constant fatigue and reduced physical capabilities since starting dialysis. One participant stated, "On dialysis days, I am so drained and in pain. I just want to lie in bed and rest; there's no energy left for anything else" (Participant 001).

**Table 3. Time on Hemodialysis and Cardiovascular Complications.**

| Participant ID | Time on Hemodialysis (Years) | Time on Hemodialysis (Months) | Cardiovascular Complications (Open Heart Surgery) |
|---|---|---|---|
| 001 | 0.5 | 6 | No |
| 002 | 6 | 72 | Yes |
| 003 | 5 | 60 | Yes |
| 004 | 5 | 60 | Yes |
| 005 | 3 | 36 | No |
| 006 | 4 | 48 | No |
| 007 | 2 | 24 | No |
| 008 | 8 | 96 | Yes |
| 009 | 5 | 60 | Yes |
| 010 | 5 | 60 | Yes |
| 011 | 2 | 24 | No |
| Total | | 546 | |
| Average | 4.14 years | 49.64 months | 60% of participants with cardiovascular complications |

One participant vividly recounted his experience with the exhausting routine of hemodialysis. He expressed a deep longing for a respite from the relentless schedule, Participant 011 shared,

I wish I could take a day off of dialysis to skip the pain and discomfort. Dialysis drains all my energy away, but I guess, as patients, we do not have the luxury to skip dialysis when we feel like it. I wish the providers could sign on for that because it will be very helpful for my mental health.

Participants' wish for a break from the routine emphasizes the psychological burden of chronic treatment. He lamented that the lack of flexibility in the treatment schedule is a significant source of stress, as it eliminates opportunities for him to manage his mental health effectively.

**Symptom management.** The transition from being relatively healthy to managing a chronic illness like CKD was overwhelming for most participants. Participants discussed the challenges they endure in adhering to a strict dietary regimen and managing their symptoms through medication and lifestyle changes. One participant shared, "Sticking to the diet and taking all the meds is tough, but I know it's necessary. It's a constant struggle, but I try my best" (Participant 003). While effective symptom management strategies, including dietary adjustments, medication adherence, and regular physical activity, are essential for improving QoL, most participants find it extremely challenging to adhere to these restrictions. Participant 003, "The symptoms can be bothersome at times; I rely on the support of my healthcare team and family to address any concerns and manage my symptoms in the best manner." Participant 004, "Apart from fatigue, I sometimes experience muscle cramps and nausea, but these are managed with medications and adjustments to my treatment plan."

Other symptoms that participants struggle to manage include difficulty falling asleep post-dialysis. A retired corporate executive participant indicated, "I struggle to fall asleep and sometimes even taking sleeping pills, I am still awake till around 4 AM" (Participant 001). However, he denied other symptoms, such as pain, fatigue, or dizziness, that are typically associated with hemodialysis.

**Information sharing.** For most participants, being a hemodialysis patient reveals significant gaps in the support and guidance provided by healthcare professionals. One critical issue is the infrequency of doctor visits and the limited scope of their advice, which often boils down to a single directive: "You need dialysis." Participant 011, a retired physician, stated,

I never knew the doctors would come so infrequently, and they would come and have just one slogan: You need dialysis. They do not encourage me to get a transplant. All I had to do was alone. They don't suggest which hospitals

to go to or what to do. So, from that administration point of view, I didn't get help. And if I didn't get help, no one else would get help because one would think I am a physician, so I should get all the help I need. I think what is lacking is a class type of education for dialysis patients. Okay, let's all sit together and see what everybody has, and I'm pretty sure somebody would bring the transplant. What is the procedure? How do you guys refer to transplants? That thing is missing.

Participants valued the care and service they received from their healthcare team. Participant 009 reflected,

Communication with my healthcare team is essential for ensuring that I fully understand my treatment schedule, procedures, and overall care plan. I am thankful for the service I received from the nurses at the dialysis center.

Also, Participant 002 noted

Even knowing that I am a physician, my healthcare team still takes the time to ensure I understand the benefits, risks, and maintenance requirements associated with each type of access, allowing me to make informed decisions about my treatment.

### Life adjustments on hemodialysis

For most participants, the life adjustments hemodialysis requires are profound and complicated. These include lifestyle changes, activity restrictions, eating habits and dietary restrictions, workforce disruptions, and financial hardships.

**Lifestyle changes.** Participants, in most cases, overhauled their daily routines to accommodate frequent medical appointments and strict hemodialysis regimens. Participants described a complete transformation in their daily schedule, focusing more on health management and less on previous hobbies or social activities. Two participants discussed how they were forced to retire from the charity work they were doing in their native countries, helping low-income families to access healthcare and attain food security. Even though they enjoy charity work, they had to reassign their administrative responsibilities due to stage 5 CKD restrictions. One participant stated, "I had to step back from my charity work. It was something I loved, but now my health comes first" (Participant 011). In terms of dietary changes, Participant 004 stated, "Since being diagnosed with CKD, my life has changed significantly. I've had to make adjustments to my diet, lifestyle, and daily routine in order to stay away from the hospital." Participant 005 echoed a similar experience and shared: "Managing my health condition requires careful attention to my diet, fluid intake, medication regimen, adherence to the rules, and avoiding community gatherings."

**Activity restrictions.** Most participants had a previously active life; however, after being diagnosed, they faced significant limitations in physical activities due to fatigue and discomfort associated with dialysis. One participant stated, "I used to be very active, but now I find myself too tired to do the things I once loved. Dialysis takes a toll on my body" (Participant 006). Participant 005 stated, "The physical symptoms and limitations associated with CKD and undergoing hemodialysis treatments have made everyday tasks more challenging." Furthermore, Participant 003 shared, "Dialysis treatments require several hours multiple times a week, which limits my flexibility in scheduling other activities, and I can't do any on dialysis day; it's tough."

**Dietary restrictions.** For most participants, managing CKD involves adhering to strict dietary guidelines to prevent complications and improve QoL. Dietary restrictions were the most frustrating limitations of hemodialysis for most participants. Participants discussed the challenge of following a restricted diet, particularly avoiding favorite foods high in potassium or phosphorus. Four participants dueled on how much they missed their favorite native dishes like biriyani, samosa, and korma. They cannot enjoy these foods anymore due to their high sodium and phosphorus levels, which harm their condition. One participant noted, "I miss my favorite biryani and samosa. It's hard to enjoy meals when you have to

be so careful about salt" (Participant 002). Another participant shared, "Fluid restrictions associated with hemodialysis are challenging for me, especially in social settings or during family gatherings where food and beverages are often abundant; it's difficult to avoid delicious food when you are served," he smiled (Participant 011).

According to most participant narratives, they had to make significant changes in eating habits to manage CKD effectively. Participants mentioned that shifting from high-sodium and high-sugar foods to a more kidney-friendly diet, which is crucial for controlling blood pressure and maintaining kidney function, is still a struggle. One participant indicated, "I've had to learn to eat differently; it's tough but necessary to keep my kidneys as healthy as possible" (Participant 003). Participant 004, "There are traditional foods that I enjoy, but I've had to make adjustments to my diet due to CKD and hemodialysis." Participant 010, "However, I have learned to prioritize my health and well-being by making conscious choices about my fluid intake, seeking alternatives to traditional native dishes that are lower in sodium and fluids, and communicating my needs to family members and friends."

**Workforce disruptions.** Professional changes are often inevitable for hemodialysis patients. One participant had to resign from his job as a construction technician due to the physical demands of his work and the debilitating fatigue caused by dialysis. Another participant, a physician, explains how he lost his career after he contracted COVID-19 and, subsequently, CKD. He shared, "I had to give up my career and many personal freedoms; it's been a tough adjustment" (Participant 011). Participant 010, "As a former ER nurse, I have always been accustomed to a fast-paced and physically demanding work environment, but my health condition has necessitated adjustments to my daily routine and responsibilities."

Maintaining a work-life balance became increasingly challenging for most participants. One participant, a former executive of a multiple billion-dollar software corporation, explained the events leading to his CKD diagnosis, which, according to him, is due to the lack of work-life balance. He narrated how he used to manage a vast team across multiple countries, including Japan, China, Israel, and Switzerland. His role required extensive monthly travel, often visiting several countries within short periods. These trips involved meeting high-profile clients, including national leaders and corporate presidents, which added to the job's stress. According to the participant, extensive travel and demanding job responsibilities led to his poor dietary habits and irregular schedules, exacerbating his pre-existing diabetes and leading to uncontrolled hypertension. These conditions were further aggravated by his reliance on ibuprofen for headaches, which eventually contributed to kidney damage. The cumulative effect of these factors resulted in a diagnosis of CKD following a severe COVID-19 infection that required intensive care. Participant 001 shared,

My job was demanding, and it took a toll on my health. I never thought it would lead to something as serious as CKD; I advise all young professionals to adhere to a work-life balance. Usually, when you are young, you do not realize the importance of good health until you lose it".

**Financial hardships.** For some participants, the cost of managing chronic illness and undergoing hemodialysis has strained their finances, affecting their ability to afford the new lifestyle. One participant shared, "The financial burden of this disease is overwhelming. I'm constantly worried about how to afford to manage my condition, now due the intense fatigue I had to hire a home aid to take care of my household chores" (Participant 010). The diagnosis of CKD and the subsequent requirement for regular dialysis sessions forced one participant, a physician, to put his medical practice on hold. For this participant, the abrupt career interruption had far-reaching consequences, affecting his financial stability and professional identity. He stated, "I lost my main source of income after being diagnosed with CKD; I could not practice after that, it has not been easy for me" (Participant 011).

**Emotional responses.** Participants shared a common experience with depression and anxiety after receiving news of their CKD diagnosis. Emotional support from healthcare providers and families helped alleviate feelings of isolation and helplessness among participants. Depression, frustration, faith, and reflection are shared emotional responses among participants.

**Depression.** Participants shared that the loss of physical health and independence contributed to their depressive symptoms. One participant explained that his experience of losing his wife to COVID-19 shortly before starting dialysis exacerbated his emotional distress. His wife, who would have been his primary support system, was no longer there to provide emotional and practical support, intensifying his feelings of loneliness and helplessness, stating, "I was devastated when I learned about my wife's death" (Participant# 011). Participant 010 noted, "The emotional impact of CKD is profound. I struggled with anxiety and depression, but regular counseling sessions and family support have been crucial in managing these feelings." Participant 008 stated, "When I found out I had kidney disease, it was like the ground disappeared beneath my feet. It took a lot of mental adjustment to accept my new reality." Participant 002: "I am scared of the uncertainties with kidney disease, the dietary restrictions, you name them; not knowing what to expect with this disease scares me."

**Frustration.** Participants described frustration with the physical limitations imposed by CKD, such as constant fatigue and the inability to perform activities they once enjoyed. This frustration is echoed in almost all participant narratives about feeling overwhelmed by the need to rely on others despite a strong desire to maintain independence. For example, the inability to drive and the need for assistance undermined their sense of independence and self-sufficiency, contributing to emotional strain. One participant explained about him experiencing episodes of collapsing post-dialysis, highlighting the physical toll of the treatment. "After every dialysis session, I feel completely drained. One time, I even collapsed. It's incredibly frustrating not to be able to do the things I used to do" (Participant#-011). Other participants also shared a similar sentiment: Participant 006: "There were moments of deep despair and frustration, but I learned to channel my emotions through creative outlets and spiritual practices, which brought me peace." Participant 007: "It's easy to get caught up in the negative emotions that come with chronic illness. I found solace in my community and through my faith, which helped me stay positive."

**Reflection.** Living with CKD has profoundly impacted not just the body, but also the emotional and mental well-being of participants in this study. Participants reflected on life changes since their diagnosis, acknowledging family support and the strength they gained from their loved ones. Most participants described their journey from initial fear and uncertainty to acceptance and adaptation, facilitated by a better understanding of their condition and its management. One participant noted, "I've had a lot of time to think about my life and what matters most. It's been a hard journey, but I've learned to appreciate the support from my family and healthcare team" (Participant#-010). Also, Participant 009, shared, "

> Coping with a chronic illness like CKD and being dependent on dialysis can be draining and tiring, affecting my mood and motivation to participate in activities I once enjoyed. I used to enjoy cleaning around my house, cutting the grass on my lawn, and you know, going for a walk. All that is an ordeal.

Similarly, Participant 004 shared the overwhelming emotional response to receiving his diagnosis. he shared,

> The news of my diagnosis was overwhelming and unsettling. It was difficult to come to terms with the fact that my kidneys were not functioning properly. I lived in self-denial for a while before I finally accepted my new reality. I hate dialysis, I can tell you that.

For these participants, the reality of living with a chronic illness like CKD is a significant challenge, considering it undermines their sense of control over their own bodies and life. The physical limitations imposed by CKD and the necessity of ongoing hemodialysis have been overwhelming for some participants, forcing many to confront the emotional burden of this new reality. For most participants, accepting the diagnosis is not just about adapting to physical changes, but also about navigating the emotional turmoil that accompanies such a life-altering condition.

Participant 005 described the emotional challenge of living with CKD as an ongoing struggle: "Living with CKD is a constant emotional battle. Some days are better than others but having a strong support system in place helps tremendously." The variability in emotional responses reflects the unpredictable nature of chronic illnesses. According to most participants, on some days, they feel more empowered or in control, while on others, the burden of the disease feels heavier. This underscores the critical role that family, friends, and caregivers play in providing the emotional support needed to cope with the ups and downs of managing CKD.

For Participant 009, the emotional journey of living with CKD has been particularly challenging. She reflected on how her mental health was often affected by the disease. Participant 009 shared, "The journey has been emotionally taxing. I often felt depressed, but engaging in religious activities provided a sense of purpose and comfort." This participant's reflection highlights how spiritual practices, such as prayer or religious engagement, can provide emotional relief and a sense of purpose. For this participant, during these struggling times, turning to religion offered hope, comfort, and a framework for understanding and navigating the emotional challenges of living with CKD.

Participant 011 discussed the emotional turmoil he experienced upon being diagnosed with CKD but emphasized the importance of connecting with others who share similar experiences: "The diagnosis brought a wave of emotional turmoil. However, connecting with others who understand my experience has been invaluable for my mental health." This statement highlights the therapeutic value of shared experiences and community. Connecting with others who understand the challenges of living with CKD has alleviated feelings of isolation and provided emotional support that strengthens the mental health of Participant 011.

In reflecting on her mental health journey, participant 001 shared his experience with the emotional and psychological preparation required for hemodialysis. He described the mental readiness he had developed over the long process leading to dialysis. Participant 001, shared, "Yeah, I was getting ready mentally because of the long process that I was going through. I saw it coming." he explained, highlighting his anticipation of the demanding treatment ahead. Despite this preparation, the reality of the four-hour dialysis sessions posed significant challenges. "it is not fun to sit there for four hours," he admitted. To cope, he brought his computer and business phone to stay productive during these sessions. Participant 001 shared, "I always take my computer. I do my work, use my business phone, and use my laptop," illustrating how he maintained a sense of purpose and normalcy.

Having retired after 30 years, Participant 001 now works as a private consultant. This professional engagement helps him stay mentally active and avoid depression. Participant 001 shared,

Mentally, I was prepared for it. I knew that my kidneys would fail at some point, so it doesn't affect me, not depressed or any of that. However, the current consultant work that I do with governments around the world keeps my mind off my current condition.

Participant 001 went on to explain, what he has observed at the dialysis center, he shared, "Most of the people, I see, at dialysis, more than 50% of the people are immigrants, from South America, mostly young women and older adults, appeared to be sad and depressed," he noted, adding that these patients often include younger women and older adults, some with severe complications like amputations due to diabetes. Participant 001 shared, "Many of them become very depressed. I see them sitting there very depressed and don't do anything, just watching TV every day".

In contrast, he uses his time at the dialysis center to stay productive and mentally engaged. Participant 001 shared, "I don't even watch TV for one minute. I do my work because I believe we must have a purpose". He emphasized the importance of having a purpose to avoid frustration and depression. He also expressed concern about the family support systems for other patients in his dialysis center who do not have family and communal support. Participant 001 shared "I don't know what kind of family support system those people have because many of those people are brought in and taken back home by the clinic's staff. Not their families". This lack of family involvement and potential isolation exacerbates mental

health challenges for those patients. Participant 001 reflected on how fortunate he felt to have the support of his family and friends, unlike the other immigrant patients who come to the dialysis center without family support. Participant 001, shared;

> If they're living in an elderly home with nobody and no family members living with them, it's pretty tricky. I cannot imagine ever having my parents living by themselves. Now, reflecting on all these I feel very thankful to Allah and my family.

### Immigration and adjustment

For most participants, immigration status has positively impacted their QoL, providing stability and access to essential services. Professional visas and citizenship are two sub-themes that emerge with immigration.

**Professional visa.** Navigating the professional visa process has been complex and stressful for some participants, while others found it easy. One participant shared his journey of moving to the U.S. on a student visa, later transitioning to a more stable immigration status through marriage to a U.S. citizen. Initially, he faced challenges securing employment and healthcare access. He stated, "When I first arrived, it was tough. I was on a student visa, and it was hard to find a job that offered health insurance; after marrying my wife, a U.S. citizen, things improved, and I was able to get the healthcare I needed" (Participant#-003).

In contrast, Participant 001, currently a U.S. citizen, shared his experience navigating the immigration and citizenship process. His journey to citizenship was notably unique due to the strategic importance of his doctoral research to the U.S. government. This significance facilitated his immigration process, as he explained, "I didn't even have to try it. It was given to me because when I was doing my doctorate, one idea I was researching was that it was strategically important to the U.S. government for technology." Recognizing the value of his work, both his employer and the U.S. government were keen to retain him in the country. "My employer and the U.S. government wanted to ensure that I didn't leave the U.S. after my doctorate program," he noted, highlighting the seamless transition he experienced. Consequently, he was granted a green card and later took the oath to become a U.S. citizen.

**Citizenship process.** Participants described how becoming a U.S. citizen transformed their lives, allowing them employment opportunities. One participant stated, "Gaining U.S. citizenship was a game-changer for me. Life began to shine for me, you know, I had a better job and I was able to file for my children from my previous marriage to come join me in the USA." (Participant 009). Before obtaining citizenship, most participants feared job insecurity and inadequate health insurance. One participant indicated, "Having my green card and later becoming a citizen wasn't easy process, but it was worth it; it has allowed me to have access to healthcare through my employer, even though in those days I did not use health much, but it was nice to have access" (Participant 005). Participant 004: "Navigating the immigration process and obtaining citizenship was quite challenging but rewarding. It required a lot of paperwork, interviews, and patience; however, now, being a citizen, I feel a sense of belonging and security in this country."

## Discussion

The findings of this study provide critical insights into how older immigrant adults perceived the impact of living with stage 5 CKD on their QoL, particularly as it intersects with their cultural beliefs and practices. While previous research has highlighted the general challenges faced by hemodialysis patients, such as physical discomfort, mental health struggles, and dependency on medical treatments [1,4,7], this study brings to light the distinct experiences of older immigrant adults, shaped by cultural, linguistic, and socioeconomic factors.

On average, participants had been receiving hemodialysis for approximately four years (Table 3). This duration carries significant health implications. Prolonged dialysis increases the risk of cardiovascular complications, such as left ventricular hypertrophy, heart failure, and atherosclerosis [41,42], which were reported by 60% of participants (Table 3), most of

whom had undergone open-heart surgery. Extended treatment is also associated with increased infections, particularly those related to vascular access points such as arteriovenous fistulas or grafts [43,44].

Many participants described fatigue, depression, and a sense of social limitation, which they attributed to strict treatment regimens and lifestyle constraints. These symptoms are well-documented in the literature as factors that significantly reduce QoL for long-term dialysis patients [44,45]. Managing these issues is critical, as they influence not only well-being but also treatment adherence.

Prognosis among long-term dialysis patients varies depending on age, comorbid conditions, and overall baseline health [1,7]. Patients with multiple chronic illnesses, such as diabetes or hypertension, tend to experience higher mortality rates, while healthier, younger individuals generally fare better [1]. A holistic care model—one that integrates physical, emotional, and social support—can significantly improve patient outcomes [46–48].

Participants' narratives underscore the complex interplay between resilience and vulnerability in managing CKD. The study emphasizes the need for comprehensive, patient-centered care that not only addresses the physical and emotional aspects of hemodialysis but also respects cultural values. Key themes include family and social support, religious practices, independence versus dependency, health perceptions, lifestyle adjustments, and immigration-related challenges.

## Family and social support

The outcome of this study highlights the indispensable role of family and social support in managing CKD. This support system is crucial for both emotional and practical reasons. For most participants in this study, emotional support from family members helps mitigate the psychological impacts of chronic illness, offering patients a buffer against feelings of isolation, anxiety, and depression. This aligns with existing literature, which consistently emphasizes the importance of social support in improving health outcomes and the QoL for hemodialysis patients [3,48–50].

The emotional toll of CKD can be overwhelming, leading to significant psychological distress. Strong emotional support from family and friends is associated with lower levels of depression and anxiety among hemodialysis patients [49]. For most participants in this study, emotional support from their families and friends provided a sense of stability and safety, helping them cope with the stress and uncertainty of their condition. Previous studies have shown that patients who receive robust emotional support tend to have better psychological outcomes and a more positive outlook on their health [3,49,50]. Hall et al. (2020) found that having a family member present at medical appointments is particularly beneficial, as it provides emotional reassurance and helps understand and manage the treatment process.

In this study, practical assistance from family members was vital for participants in managing CKD. This includes helping with daily activities, transportation to medical appointments, and ensuring adherence to treatment schedules. According to participants in this study, such support significantly alleviated the physical burden of CKD, allowing them to focus more on their recovery and less on the logistical challenges of managing their condition. Previous studies confirmed that practical support ensures that patients can maintain a semblance of normalcy in their daily lives, which is crucial for their overall well-being [3,48,50,51]. For most participants in this study, beyond the family, community support also plays a crucial role in their well-being, and they expressed gratitude for the support they receive from their communities. Community support networks can provide additional layers of emotional and practical assistance [48,50].

Community involvement provided most participants with a sense of belonging and purpose, which is essential for mental health and overall well-being in patients with chronic illnesses [48,50,52]. According to participants, the community support systems helped them create an environment where they felt understood and supported, reducing the sense of isolation that often accompanies chronic illness. Comprehensive support from family, friends, and the community directly impacts health outcomes [50]. Patients with strong social support systems are more likely to adhere to their treatment plans, attend medical appointments regularly, and engage in healthy behaviors [50,51]. This adherence is critical for managing CKD effectively and preventing complications. Moreover, the emotional and psychological benefits

of social support can lead to improved physical health, as mental well-being is closely linked to physical health outcomes [51].

## Cultural and religious practices

Cultural heritage and religious identity profoundly influenced how participants in this study perceived and managed their chronic illness. These frameworks were not merely contextual—they shaped beliefs about suffering, healing, and endurance. For many, religious and cultural practices served as essential tools for psychological coping and emotional resilience, offering meaning and comfort amid the disruptions caused by CKD. Participants often described religious rituals such as daily prayers, mosque attendance, and spiritual reflection as sources of inner strength, social connection, and hope. These findings are aligned with research indicating that spirituality and religiosity can mitigate psychological distress, strengthen coping mechanisms, and enhance overall well-being among chronically ill individuals [53–55].

A particularly compelling example of the intersection between health and faith was the practice of fasting during Ramadan, discussed by several Muslim participants. Fasting during Ramadan is a religious obligation with deep spiritual significance, often tied to personal identity, community belonging, and moral discipline. However, for individuals on hemodialysis, this practice introduces considerable medical risk due to fluid and dietary restrictions, risk of hypotension, and metabolic instability. Participants shared deeply personal accounts of the emotional and ethical dilemmas they faced,

choosing between spiritual fulfillment and health preservation. Some continued to fast despite medical warnings, guided by faith, familial expectations, or fear of religious judgment. Others struggled with guilt or disappointment when advised to abstain from fasting, indicating a sense of loss not only of religious observance but of communal participation.

This nuanced conflict illustrates how religious observance can both support and complicate chronic illness management. It also underscores the need for culturally informed clinical dialogue—conversations where patients feel safe discussing religious commitments and where providers offer guidance rooted in respect, empathy, and medical ethics. As Elliott et al. (2019) argue, healthcare providers should be prepared to navigate such tensions collaboratively, helping patients explore safe religious accommodations (e.g., alternative forms of devotion or partial fasting strategies) that honor both their spiritual values and health priorities.

In addition to religious rituals, participants also derived strength from cultural traditions grounded in their countries of origin. For example, two participants shared how reliance on culturally specific foods, extended family caregiving roles, and community rituals shaped their self-care practices and expectations of recovery. These cultural frameworks offered practical coping strategies and emotional resilience, especially in contexts where participants felt culturally or linguistically marginalized within the U.S. healthcare system [54,56]. In many cases, cultural norms about intergenerational caregiving or communal decision-making shaped how participants viewed treatment adherence and support systems.

Given these insights, it is essential that healthcare providers adopt a holistic and culturally competent approach to care. This involves more than linguistic translation—it requires understanding how religious identity, cultural values, and communal expectations intersect with medical decision-making. As Harding and Oetzel [57] and Chatters et al. [58] emphasize, culturally attuned care that respects patients' belief systems enhances satisfaction, improves adherence, and fosters stronger therapeutic relationships. By openly acknowledging religious and cultural rituals—such as Ramadan fasting—clinicians can co-create personalized treatment strategies that align medical recommendations with patients' spiritual and cultural goals. This integrative care model not only supports biomedical outcomes but also fosters the sense of dignity, purpose, and continuity that defines a meaningful QoL for older immigrant adults with CKD [59,60].

## Balancing independence and dependence

Participants in this study expressed a strong desire to maintain their independence while acknowledging the necessity of assistance, particularly as their health conditions deteriorate. In this study, the struggle to balance independence and dependence is a recurring theme in most participant narratives. Initially reluctant to rely on others, participants eventually

recognized the importance of accepting help from family and friends. Tsay and Hung (2015) found that social support significantly predicts better health outcomes and improved QoL for hemodialysis patients. Balancing independence and dependence is critical for self-esteem and mental health [61]. Shahgholian and Yousefi (2019) emphasize that supporting autonomy in hemodialysis patients significantly improves their QoL. The ability to manage daily activities independently is closely linked to better QoL and reduced feelings of helplessness [62–64]. Similarly, Entwistle et al. (2010) and Cook & Jassal (2008) studies found that preserving autonomy allows patients to maintain their dignity and self-worth, which are essential for psychological well-being. Four participants in this study illustrated this sentiment by describing how they managed errands and personal affairs independently, even though they recognized the need to accept help when necessary.

Participants described the emotional challenges of relying on family for transportation and daily tasks; however, despite these challenges, they strived to maintain a sense of normalcy and well-being amidst the adjustments required by CKD. Kim & Cho (2021) confirm that strong social support networks are essential for coping with hemodialysis's physical and emotional demands. Patients who accept help tend to experience lower levels of depression and anxiety and better adherence to treatment regimens [64,65]. In this study, most participants have adapted well to their treatment routines and strive to adhere strictly to their healthcare regimens, recognizing their importance in maintaining health. This active participation is crucial for patient empowerment and enhances their ability to manage their health effectively [62,65].

## Health and illness perceptions

Understanding the QoL among older immigrant adults with stage 5 CKD necessitates a nuanced exploration of how cultural values shape health perceptions and treatment engagement [35]. Many participants viewed their illness not only as a biomedical condition but also as a disruption of spiritual balance or a reflection of personal or familial shortcomings. In several cultures represented in this study, chronic illness was interpreted as divine will, ancestral imbalance, or the consequence of moral failing, leading to feelings of shame, guilt, or spiritual distress. These interpretations are deeply rooted and are not easily addressed through conventional clinical approaches.

Participants emphasized the need for culturally sensitive and emotionally attuned communication from healthcare providers. When such dialogue was present, it helped bridge the gap between clinical knowledge and personal belief systems, ultimately enhancing comprehension and adherence. For example, a few participants shared that when providers acknowledged their spiritual beliefs or offered alternative explanations compatible with cultural norms, they felt more respected and empowered to engage in care. In contrast, lack of explanation, especially regarding transplant eligibility, dialysis duration, or dietary restrictions, resulted in disengagement, mistrust, or fatalistic thinking. These communication breakdowns are often compounded by differing cultural expectations around authority, disclosure, and decision-making in medical contexts [61,66].

The moment of diagnosis was particularly distressing. Some participants reported receiving life-altering information with little support, while others interpreted the diagnosis within cultural frameworks that equate illness with spiritual vulnerability or divine judgment. These interpretations sometimes delay emotional processing and acceptance of treatment. As supported by research, compassionate and culturally informed disclosure at diagnosis can foster trust, reduce fear, and promote sustained treatment engagement [67,68].

Cultural expectations also shaped participants' views on dialysis. For some, dialysis was a life-saving opportunity that aligned with religious values of perseverance. For others, particularly those from traditions that prioritize natural aging or holistic wellness, dialysis was seen as a burdensome prolongation of suffering. One participant described it as "a necessary evil." These differing views emphasize the importance of culturally inclusive discussions about prognosis and treatment goals.

In addition to emotional and spiritual concerns, financial insecurity significantly influenced patient experiences. Worries about insurance, medication costs, and burdening family members often exacerbate stress and affect adherence. In many cultural contexts where elder autonomy is highly prized, relying on others for transport, financial support, or basic care undermined self-worth and added to psychological distress [63,69].

Lastly, physical symptoms such as fatigue and pain were not only medical issues but also carried cultural meaning. In cultures that associate aging with productivity or wisdom, needing help with daily tasks led to a sense of humiliation [70]. Participants noted that their physical decline made them feel like a burden, especially in households where multigenerational caregiving is not traditionally expected.

To address these layered challenges, CKD management must incorporate culturally competent strategies that go beyond clinical symptom control [69,70]. These include individualized care planning, culturally adapted behavioral therapies, and respectful acknowledgment of spiritual and cultural interpretations of illness. By honoring these dimensions, providers can deliver more equitable, compassionate, and effective care to immigrant populations living with CKD [70,71].

## COVID-19 as a trigger for CKD

The COVID-19 pandemic has profoundly impacted global health, not only through direct viral infections but also by exacerbating preexisting conditions and contributing to new chronic illnesses [72]. Among the participants in this study, three reported that their COVID-19 infections directly precipitated their CKD diagnosis and need for dialysis. Similarly, three others with a history of diabetes described severe COVID-19 episodes that led to ICU admissions and ultimately to end-stage renal disease. These accounts reflect emerging clinical findings that COVID-19 can severely impair kidney function and precipitate long-term renal complications [73,74].

Several studies now confirm the strong association between COVID-19 and acute kidney injury (AKI), particularly among hospitalized and critically ill patients. For example, Hirsch et al. (2020) found that AKI occurs in up to 36.6% of hospitalized COVID-19 cases and in over 50% of ICU patients. AKI, once considered a transient complication, has increasingly been shown to have lasting consequences, with many patients failing to regain baseline kidney function [75,76].

In this study, participants described the abrupt onset of renal failure as both physically and emotionally destabilizing. Their experiences mirror clinical data showing that approximately 30% of COVID-19 patients who develop AKI do not recover full kidney function by discharge, with 5–10% requiring long-term dialysis [74]. These risks are magnified in individuals with underlying conditions such as diabetes and hypertension, comorbidities prevalent among the study's participants [76].

The progression from COVID-19-induced AKI to CKD highlights the need for integrated care models that include early nephrology consultation, especially for patients admitted to ICUs with known risk factors. The inflammatory and thrombotic cascades triggered by COVID-19 can accelerate renal damage, emphasizing the urgency of post-infection renal surveillance [74].

## Life adjustments to hemodialysis

Participants in this study described profound limitations in physical functioning due to fatigue and discomfort related to hemodialysis. These physical constraints significantly restricted their ability to engage in previously enjoyed activities, contributing to frustration and diminished life satisfaction. Griva et al. (2011) confirm that symptoms like fatigue and muscle weakness are common among hemodialysis patients and can severely impact QoL. This loss of function often led to social withdrawal, compounding the emotional burden and decreasing overall well-being [77].

One of the most challenging aspects reported by participants was the adherence to strict dietary guidelines, particularly the restriction of foods high in potassium and phosphorus. Participants felt that these limitations not only affected their physical health but also disrupted social and cultural traditions related to food. Despite recognizing the medical necessity of these dietary changes, many found them isolating and emotionally taxing. Kalantar-Zadeh et al. (2015) and Cardol et al (2022) highlighted how such restrictions, while critical for disease management, can reduce social engagement and mental health.

The cumulative physical and emotional toll of long-term dialysis treatment underscores the importance of holistic care that incorporates physical, emotional, and social dimensions [75,78,79]. Participants had to substantially adjust their daily routines, hobbies, and social engagements to accommodate the demanding treatment schedule. One participant

poignantly remarked, "I wish I could take a day off dialysis," illustrating the relentlessness of the regimen. Kalantar-Zadeh et al. (2020) similarly note that the time-intensive nature of dialysis can significantly encroach on patients' personal lives.

Professional adjustments were also necessary for many. Due to severe fatigue and physical strain, some participants retired early or reduced their working hours. Employment disruptions frequently led to financial insecurity and increased stress [80,81]. This aligns with Kutner et al. (2023), who reported high rates of job loss and reduced work capacity among hemodialysis patients. Even for those who had retired, the desire to remain professionally active persisted, though many acknowledged that continuing to work was not realistically sustainable given their health status.

Work-life balance was a persistent challenge. The need for flexible work environments and supportive policies was emphasized by those who continued working. Wong et al. (2019) found that employer flexibility is vital in helping patients manage treatment demands alongside their professional responsibilities, which in turn improves QoL.

Financial hardship emerged as a critical concern. The cumulative costs of chronic illness, medications, transportation, and lifestyle adjustments strained many participants' finances. For older immigrant adults with CKD, access to financial assistance and social services is not only helpful but often essential. Studies by Walker-Pow (2024) and Hamilton et al. (2013) confirm that financial insecurity undermines healthcare access and heightens psychological distress, ultimately diminishing QoL in this population.

## Emotional responses

Depression, frustration, and reflection emerged as central emotional responses among older immigrant adults undergoing hemodialysis. Addressing these dimensions is essential to delivering comprehensive, human-centered care. Participants consistently reported that the loss of physical health and independence, coupled with the major life adjustments dialysis demands, contributed significantly to depressive symptoms—an experience well-documented in the literature, with prevalence estimates ranging from 20% to 30% [82]. Depression in this population is closely tied to reduced QoL, poorer treatment outcomes, and increased mortality [83,84,85]. Many participants described receiving their diagnosis as a devastating moment, often accompanied by fear and despair—a pattern consistent with findings that highlight the psychological toll of CKD and its treatment regimen [77,85].

Participants also expressed frustration related to their diminished autonomy and reliance on others. Challenges such as chronic fatigue, inability to drive, and post-dialysis exhaustion disrupted daily routines and undermined self-sufficiency. This emotional strain reflects the broader impact of CKD on personal freedom and identity, especially among those accustomed to independence [86]. One participant described collapsing after dialysis sessions—an event that magnified his physical vulnerability and emotional distress. The repetitive nature of treatment and the intrusive role it plays in everyday life can generate frustration and disengagement, further reducing QoL [82].

Spirituality emerged as a vital source of resilience for many participants, providing emotional grounding and a framework for coping. Personal prayer, religious rituals, and a sense of divine connection helped patients find solace amid uncertainty. These findings align with research demonstrating that religiosity and spirituality enhance psychological well-being, reduce anxiety and depression, and improve coping in chronic illness contexts [46,87]. For many, faith also fostered a sense of community support, which buffered against isolation and encouraged emotional stability. Integrating patients' spiritual needs into clinical care has been shown to strengthen therapeutic relationships and improve mental health outcomes [49,52,88].

Reflection played a pivotal role in helping participants process their illness and reconstruct meaning. Many described a journey from initial fear to eventual adaptation, facilitated by family support and increased understanding of CKD management. This reflective process enabled a reorientation of self and promoted emotional growth. Research affirms the therapeutic value of reflection in chronic illness, suggesting it fosters resilience and supports identity integration [89]. Promoting

reflective practices through counseling or peer groups can therefore enhance QoL by supporting patients in making sense of their illness journey and reclaiming agency [81,89].

## Immigration adjustment

Achieving a stable immigration status substantially improves healthcare access and enhances QoL for most participants. Although the path to citizenship involves complex and often stressful bureaucratic procedures, most participants in this study viewed it as a worthwhile investment. Gaining legal residency enabled QoL through improved healthcare access and adherence to treatment regimens—findings supported by literature showing that legal stability reduces stress and promotes positive health outcomes [35].

However, the transition to life in the U.S. presented significant legal, social, and cultural hurdles. Participants reported that adjusting to unfamiliar norms while settling in the U.S. contributed to psychological distress and feelings of uncertainty. These challenges underscore the need for culturally competent care that addresses not only language barriers but also patients' broader sociocultural experiences [90]. Effective healthcare delivery for immigrant populations must be inclusive of their unique needs, ensuring that cultural differences do not become obstacles to care.

Support systems emerged as a vital buffer against these multifaceted challenges. Participants consistently emphasized the importance of legal aid, social services, and emotionally supportive healthcare environments in helping them navigate the U.S. healthcare system. Research confirms that integrated support structures can mitigate the strain of immigration-related stressors and foster improved QoL among immigrant CKD patients [51,67].

Community-based interventions, such as peer support groups and culturally informed counseling, also facilitate smoother adaptation to a new cultural environment while encouraging health literacy and treatment engagement [90]. The complexity of the U.S. medical system can be overwhelming, especially for those unfamiliar with its structure, yet family and community networks, when mobilized effectively, can help bridge these gaps [91]. Collectively, these findings highlight that comprehensive, culturally attuned support is not supplemental but essential to equitable care for immigrant populations with chronic illnesses.

## Rigor and trustworthiness

This study established rigor and trustworthiness by addressing dependability, credibility, transferability, and confirmability [92]. Dependability refers to the extent to which the findings were consistent and the reliability of the research findings [92]. The analysis expert, who had no vested interest in the study outcome, was consulted on the data analysis to establish dependability. Any discrepancies regarding codes, themes, and sub-themes were resolved through critical discourse.

Credibility refers to the extent to which the findings are believable [93]. Member checks and prolonged engagement with the data were performed to establish credibility. To ensure member check-in, the primary researcher reviewed the interview transcripts with each participant via phone and sent a copy via email for feedback on their perspective of the primary researcher's interpretation of their data.

Transferability refers to the extent to which the findings could be applied to external contexts and settings [94]. To verify transferability in this study, findings were discussed and situated within an appropriate and applicable research context and a relevant theoretical framework demonstrating how they could be applied to improve practice. Also, the demographic data that was collected enhanced transferability.

Finally, confirmability refers to objectivity in the study's data collection and analytical aspects [93]. To establish confirmability in this study, an audit trail was maintained throughout the research process, containing thorough documentation of all study components and observations. This allowed other researchers to verify how themes and sub-themes were identified. These steps were expected to lead to valid findings that significantly contributed to understanding older immigrant adult patients' experiences with stage 5 CKD.

## Limitations of this study

It is important to acknowledge that this study has several limitations. The sample size is relatively small and geographically concentrated in the Mid-Hudson Valley Region of New York, which may limit the generalizability of the findings. Also, considering that many older immigrant adults on hemodialysis are non-English speakers, this study excluded this population, which has limited its generalizability. Additionally, the qualitative nature of the study relies on self-reported data, which can be subject to recall bias and social desirability bias. Being aware of these limitations is crucial for understanding the study's scope and implications.

## Study significance & clinical implications

The findings of this study highlight the critical need for healthcare systems to adopt a more holistic, culturally responsive approach to treating older immigrant adults on hemodialysis. These patients face intersecting physical, emotional, cultural, and social challenges that significantly impact their QoL. Recognizing and addressing these complexities in clinical settings can improve health outcomes and care satisfaction.

1. **Promote Cultural Competency and Tailored Communication** Healthcare providers should receive training in cultural humility and culturally responsive care. This includes learning about patients' religious and dietary practices, communication preferences, and culturally rooted illness perceptions. For example, clinicians treating Muslim patients should be equipped to counsel on safe fasting practices during Ramadan, using shared decision-making to balance religious obligations with clinical safety. Providers should also use interpreters and culturally appropriate educational materials to ensure patients fully understand their diagnoses and treatment options.

2. **Strengthen Psychosocial Support Systems** Chronic illness management requires more than clinical intervention. Providers should routinely assess patients' mental health, screen for depression and anxiety, and refer to culturally appropriate mental health services when needed. Peer support groups, particularly those designed for immigrant or minority populations, can offer communal spaces for emotional expression, reduce isolation, and promote resilience.

3. **Integrate Family and Community Support in Care Planning** This study shows that family involvement is essential for treatment adherence and psychological well-being. Clinicians should encourage the participation of family members in care discussions and offer flexible appointment scheduling to accommodate caregivers. Community health workers or cultural navigators can also serve as liaisons between providers and patients, bridging cultural and linguistic gaps.

4. **Address Social Determinants of Health** Older immigrant adults often face structural barriers, such as limited insurance coverage, food insecurity, or unstable immigration status, that influence their ability to access care. Social workers and care coordinators should be integrated into nephrology teams to help patients secure essential services, including financial assistance programs, housing support, and legal aid when appropriate.

5. **Facilitate Patient Autonomy and Shared Decision-Making** While many patients may need practical support, preserving autonomy remains vital. Clinicians should involve patients in their care plans, respecting their preferences and values. Options for home dialysis, transportation support, or flexible dialysis scheduling should be discussed openly to empower patients and reduce treatment-related disruptions.

6. **Provide Post-COVID-19 Renal Monitoring** Given that several participants developed CKD after severe COVID-19 infection, clinicians must be vigilant in monitoring renal function in high-risk populations post-COVID. Early screening and intervention can prevent progression to dialysis-dependent disease. Providers should also be aware of the emotional trauma associated with sudden illness onset, particularly in cases linked to pandemic-related hospitalization and isolation.

7. **Prioritize Continuity of Care and Long-Term Planning** Managing chronic kidney disease in older immigrant adults is a long-term commitment. Healthcare teams should emphasize continuity of care across transitions, from outpatient dialysis units to hospital stays, ensuring patients do not fall through the cracks. Regular QoL assessments and care plan updates should be standard practice to reflect patients' evolving needs.

## Future research directions

Future research should explore the experiences of a more diverse and larger sample of older immigrant adults with CKD across different regions and include patients who speak other than English to enhance the transferability of the findings. Longitudinal studies could provide deeper insights into how patients' experiences and QoL evolve over time. Additionally, research should investigate the effectiveness of specific interventions, such as structured educational programs and culturally tailored care plans, in improving outcomes for this population.

## Conclusion

This study illuminates the complex realities faced by older immigrant adults living with stage 5 CKD on hemodialysis. It reveals how cultural identity, family dynamics, emotional resilience, and structural barriers jointly shape their QoL. While the clinical burden of prolonged dialysis is widely acknowledged, these findings emphasize that the lived experience of chronic illness is profoundly shaped by social context and personal meaning, factors too often overlooked in biomedical care.

Participants' stories reflect a continual negotiation between independence and reliance, cultural preservation and clinical compliance, spiritual beliefs and medical imperatives. These tensions, simultaneously burdensome and empowering, highlight the inadequacy of purely clinical frameworks in addressing chronic illness. Instead, they call for a deeper, more inclusive understanding of care that recognizes both struggle and strength as inherent parts of the patient experience.

Significantly, this study advances the argument for culturally responsive and patient-centered care models. QoL, as expressed by participants, extends beyond symptom control to include the ability to engage in culturally meaningful practices, such as preparing traditional foods, participating in spiritual rituals, and receiving familial support. These elements are not peripheral; they are central to emotional well-being and treatment adherence. Integrating them into care planning fosters dignity, trust, and sustained engagement in treatment.

Family support emerged as a cornerstone of emotional and practical stability, buffering the psychological toll of CKD by enhancing a sense of belonging and reducing isolation. Participants also emphasized the role of spirituality as a personal reservoir of strength, an insight supported by research linking religious practices to greater emotional resilience and reduced distress in chronic illness. Incorporating these spiritual values into clinical encounters can improve patient satisfaction and deepen therapeutic alliances.

The study further reveals a nuanced tension between autonomy and dependence. While participants sought to maintain self-sufficiency, the progressive nature of CKD necessitated increasing reliance on others. This duality often produces emotional strain. For healthcare providers and caregivers, the challenge lies in supporting patients' independence while offering needed assistance, striking a balance that affirms agency and reduces the risk of emotional erosion.

Another critical insight concerns the lack of accessible, culturally appropriate education about treatment options, particularly kidney transplantation. Several participants expressed confusion or misinformation about eligibility or availability, often the result of systemic communication gaps. Addressing these gaps through structured, peer-informed education programs can empower patients with knowledge, support informed decision-making, and help reduce disparities in access to advanced treatments.

In sum, this study calls for a reimagining of chronic illness care, one that embraces the full humanity of patients. Delivering holistic, culturally attuned care that respects spiritual, familial, and social dimensions is not merely compassionate is essential to improving outcomes and equity for older immigrant adults with CKD. Future research should build on these

insights with broader and more diverse samples, while health systems must evolve to deliver care that reflects the pluralistic realities of the patients they serve.

## Supporting information

**Appendix A.  Interview Guide.**
(DOCX)

**Appendix B.  Cognitive Assessment Tool Short Blessed Test (SBT)1.**
(DOCX)

**Appendix C.  Study Recruitment Flyer.**
(DOCX)

**Appendix D.  Consent to Participate in a Research Study.**
(DOCX)

## Acknowledgments

I want to express my deepest gratitude to the incredible individuals who have supported me throughout the Doctor of Health Science (DHSc) program and the dissertation journey. First and foremost, I extend my heartfelt thanks to my esteemed committee members: Dr. Elizabeth Moore, Dr. Laura Santurri, and Dr. Tiffany Washington. Your tireless efforts, invaluable expertise, and unwavering encouragement have been instrumental in guiding me toward the completion of this project. Dr. Moore, as my committee chair, your leadership and mentorship helped me stay focused and driven. Dr. Santurri, your insightful feedback and thoughtful critiques have elevated my work to new heights. Dr. Washington, your perspectives have enriched my research and allowed me to consider broader dimensions of this study. From the depths of my heart, I thank you all.

## Author contributions

**Conceptualization:** Demba Keita.

**Formal analysis:** Demba Keita.

**Methodology:** Demba Keita.

**Writing – original draft:** Demba Keita.

**Writing – review & editing:** Demba Keita.

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
