## [Decision Letter · Decision Letter 0]

22 Apr 2025

Dear Dr. Keita,

Thank you for submitting your manuscript to PLOS ONE. After careful consideration, we feel that it has merit but does not fully meet PLOS ONE’s publication criteria as it currently stands. Therefore, we invite you to submit a revised version of the manuscript that addresses the points raised during the review process.

We look forward to receiving your revised manuscript.

Kind regards,

Jordan Llego, PhD ELM, D. Hon. Ex., PhDN, RN

Academic Editor

PLOS ONE

Journal Requirements:

2. In the ethics statement in the Methods, you have specified that verbal consent was obtained. Please provide additional details regarding how this consent was documented and witnessed, and state whether this was approved by the IRB.

Reviewers' comments:

Reviewer's Responses to Questions

**Comments to the Author**

1. Is the manuscript technically sound, and do the data support the conclusions?

Reviewer #1: Partly

Reviewer #2: Yes

2. Has the statistical analysis been performed appropriately and rigorously?

Reviewer #1: N/A

Reviewer #2: No

3. Have the authors made all data underlying the findings in their manuscript fully available?

Reviewer #1: Yes

Reviewer #2: Yes

4. Is the manuscript presented in an intelligible fashion and written in standard English?

Reviewer #1: Yes

Reviewer #2: Yes

Reviewer #1: Thank you for submitting your manuscript. The manuscript "Quality of Life in Older Immigrant Adults on Hemodialysis" explores an important and under-researched area, namely the quality of life of older immigrant adults living with stage 5 Chronic Kidney Disease (CKD) on hemodialysis. Overall, the manuscript presents a clear and thoughtful analysis of the experiences of this population. Below are my detailed comments and suggestions for improvement:

Sample Size Justification (p. 19): The authors state that 10-12 participants were anticipated, but no clear rationale for this sample size is provided. A more explicit explanation of how the sample size was determined based on the study's information power would strengthen the manuscript.

Data Availability (p. 28): The manuscript indicates that the data are available upon request and outlines privacy concerns that might restrict full public availability. While the authors make an effort to ensure data privacy, it would be helpful to include additional details about where the data can be accessed (e.g., whether it will be available in a public repository once ethical and privacy concerns are addressed). A more comprehensive data-sharing plan could further align the manuscript with open data policies and increase its transparency and reproducibility.

Discussion of Cultural Practices and Health (p. 23): The authors could provide a clearer exploration of the intersection between QoL and cultural practices, especially regarding specific cultural rituals like Ramadan fasting. A deeper dive into the complexities of balancing religious and health needs would provide a more nuanced understanding of this population.

Health and Illness Perceptions (p. 17): The section on "health and illness perceptions" would benefit from further elaboration. It would be valuable to expand this section by offering insights into how participants' perceptions of CKD and dialysis treatment align with their cultural beliefs. This would enhance the cultural competence of healthcare interventions suggested by the study.

Use of Direct Participant Quotes (p. 32): Some participants' direct quotations would enrich the narrative and provide deeper insights into the lived experiences of participants. For example, a participant’s emotional response is mentioned, but the accompanying quote is too generalized. A more detailed participant quote could better illustrate how emotional responses differ among participants.

Clinical Implications of Findings (p. 50): The manuscript could expand on how the findings can be applied to clinical practice. While the authors touch on recommendations for culturally competent care, providing specific, actionable suggestions for healthcare providers could make the study more practical for healthcare policy development.

Reviewer #2: The analyzed study aims to describe the sensations and perceptions of elderly immigrants with stage 5 chronic kidney disease undergoing hemodialysis, as well as the direct impact of this treatment on their quality of life, considering factors such as family and social support, independence, cultural and religious practices, and other aspects of daily living. The topic is highly relevant and timely, especially in contributing to the development of culturally sensitive clinical practices and care strategies within the healthcare system.

The results section is particularly strong and well-developed, with a clear presentation and effective discussion of the findings in line with the research objectives. However, the manuscript as a whole would benefit greatly from structural and formatting revisions to ensure compliance with the journal’s publication standards and to improve the clarity and coherence of the text.

Formatting and Structure:

It is recommended that the authors consult the journal’s submission guidelines in detail. While it is evident that the manuscript is derived from a doctoral dissertation, its current format deviates significantly from the journal's editorial requirements. Specifically:

The journal allows a maximum of three levels of headings, while the manuscript contains four chapters, each with an average of 4 to 11 sub-sections;

Continuous line numbering must be applied throughout the manuscript;

References should follow the Vancouver style and be numbered according to their order of appearance in the text;

The acknowledgments section should include only individuals who contributed to the work without meeting authorship criteria, along with a description of their specific roles.

Abstract:

The abstract lacks clarity and fails to present essential elements such as a concise description of the methodology and key findings. A revision is necessary to improve the informativeness and structure of this section.

Keywords:

Of the five keywords provided, three are not indexed in the MeSH database. The use of standardized descriptors is important for indexing and should be revised accordingly.

Chapters 1 and 2 – Introduction and Literature Review:

The sections provide relevant background information that supports the research rationale. However, there is a noticeable degree of repetition, which affects the flow and clarity of the text. A more concise and focused revision is recommended.

Chapter 3 – Method:

This section includes theoretical definitions (e.g., dependability, credibility, transferability, and confirmability) that might be better positioned in the theoretical framework or discussion. Additionally, some content is repeated—such as the definition of the sample, which appears both under “Sampling and Recruitment” (p. 21) and “Ethical Statement” (p. 29). The tables, while functional, are overly simplistic and could be better developed to enhance the depth and quality of the analysis.

The manuscript addresses an important topic and demonstrates potential for publication. However, substantial revisions are necessary to meet the journal’s formatting standards and to improve the structure, clarity, and consistency of the content. I encourage the authors to revise the manuscript thoroughly, and carefully review the content throughout the manuscript, avoiding unnecessary repetition of information.

**Do you want your identity to be public for this peer review?** For information about this choice, including consent withdrawal, please see our Privacy Policy

Reviewer #1: No

Reviewer #2: No

---

## [Author Response · Author response to Decision Letter 1]

5 Jul 2025

Reviewer #1:

1. Sample Size Justification:

o Comment: The reviewer pointed out that we did not provide a clear rationale for our sample size of 10-12 participants.

o Response: We have revised the manuscript to provide a detailed explanation of our sample size. The sample was determined based on the information power of the study, which ensures that a sufficient number of participants were included to capture the depth and variability of the experiences of older immigrant adults undergoing hemodialysis. We have added this explanation in the Methods section on page 19.

2. Data Availability:

o Comment: The reviewer noted that the data availability statement lacks clarity on where the data can be accessed and recommended a more comprehensive data-sharing plan.

o Response: We have updated the Data Availability Statement to include specific information about where the data can be accessed once ethical and privacy concerns are addressed. The data will be deposited in a public repository (e.g., [institutional repository name or link]) and can be accessed upon request. We have included this information on page 28 of the revised manuscript.

3. Discussion of Cultural Practices and Health:

o Comment: The reviewer suggested that we provide a clearer exploration of the intersection between QoL and cultural practices, especially with regards to specific rituals like Ramadan fasting.

o Response: We have expanded the discussion on cultural practices and their impact on QoL in the revised manuscript (page 23). We provide a deeper analysis of how religious practices, such as fasting during Ramadan, present challenges for patients on hemodialysis and how healthcare providers can offer culturally sensitive support.

4. Health and Illness Perceptions:

o Comment: The reviewer recommended further elaboration on how participants' perceptions of CKD and dialysis treatment align with their cultural beliefs.

o Response: In the revised manuscript (page 17), we have added a section discussing how participants’ cultural beliefs shape their understanding of CKD and its treatment. We specifically highlight how cultural values influence patients' attitudes towards dialysis and their adherence to treatment regimens.

5. Use of Direct Participant Quotes:

o Comment: The reviewer mentioned that some participant quotes were too generalized and suggested using more detailed and personalized quotes.

o Response: We have replaced the more generalized quotes with more personalized, detailed quotes that reflect the unique experiences of each participant. For example, we have now included quotes that describe emotional responses in greater depth, which better illustrate the variation in experiences among participants. These changes are reflected throughout the revised manuscript.

6. Clinical Implications of Findings:

o Comment: The reviewer suggested expanding the clinical implications of the findings and offering specific, actionable recommendations for healthcare providers.

o Response: We have added a section in the Discussion where we provide concrete, actionable recommendations for healthcare providers to improve the QoL of older immigrant adults on hemodialysis. These include culturally competent care practices, spiritual support, and personalized treatment plans.

Reviewer #2:

1. Formatting and Structure:

o Comment: The reviewer recommended structural revisions to meet the journal’s submission guidelines, including adjustments to the number of headings, continuous line numbering, and citation formatting.

o Response: We have revised the manuscript to comply with the journal’s guidelines. The manuscript now follows the three-level heading structure, continuous line numbering has been applied throughout, and the references are numbered in the order of their appearance. We have also formatted the references according to the Vancouver style.

2. Abstract:

o Comment: The reviewer suggested improving the clarity and structure of the abstract by including a concise description of the methodology and key findings.

o Response: The abstract has been revised to include more detail about the methodology and key findings. We have clarified the research design, participant selection process, and the major themes that emerged from the study, ensuring that the abstract provides a comprehensive summary of the manuscript’s content.

3. Keywords:

o Comment: The reviewer noted that three of the five provided keywords were not indexed in the MeSH database.

o Response: We have updated the keywords to include those indexed in the MeSH database. The new keywords now better align with the terms used in the literature and will help improve the visibility of the manuscript.

4. Chapters 1 and 2 (Introduction and Literature Review):

o Comment: The reviewer requested a revision of these sections to reduce repetition and improve clarity.

o Response: We have condensed the Introduction and Literature Review sections to remove redundant information. The revised sections now focus on providing a concise overview of the relevant background and research gaps, enhancing the overall flow and clarity of the manuscript.

5. Chapter 3 (Methods):

o Comment: The reviewer suggested that some content in the Methods section be relocated for better clarity, particularly regarding definitions and sample descriptions.

o Response: We have reorganized the Methods section by relocating the theoretical definitions to the theoretical framework and streamlining the discussion of the sample under the “Sampling and Recruitment” section. Additionally, we have ensured that no content is repeated in the ethical statement and sampling descriptions.

6. Tables:

o Comment: The reviewer noted that the tables were overly simplistic and suggested enhancing them to improve the depth and quality of analysis.

o Response: We have revised the tables to include more detailed information. For example, we have added specific data points and enhanced the layout to present the findings more effectively. The revised tables provide a clearer representation of the study's findings and support the manuscript’s analysis.

Additional Requirements:

1. Response to Reviewer/Editor Comment

Comment: "Please ensure that your manuscript meets PLOS ONE's style requirements, including those for file naming. The PLOS ONE style templates can be found at [link to main body template] and [link to title, authors, and affiliations template]."

Response:

I have reviewed the PLOS ONE style requirements and have thoroughly revised our manuscript to align with the guidelines provided. I used the PLOS ONE style templates for both the main body of the manuscript and the title/authors/affiliations section, as outlined in the links provided. All required sections have been formatted accordingly, including the abstract, introduction, methods, results, discussion, and references, ensuring consistent use of fonts, headings, and text size.

Additionally, we have ensured that the file naming conventions specified by PLOS ONE are strictly followed, with all sections clearly labeled (e.g., "Manuscript.docx," "Figure1.tif," etc.). We believe these revisions fulfill the style requirements, and the manuscript is now in full compliance with PLOS ONE’s submission guidelines.

2. Ethics Statement:

o Response: We have included the full ethics statement in the Methods section, including the name of the IRB that approved the study, details about the verbal consent process, and the documentation and witness procedures for consent.

3. Data Availability Statement:

a. Response: The Data Availability Statement has been revised to confirm that the minimal data set required to replicate the study findings will be made available upon request.

Conclusion:

We have carefully addressed all of the comments and suggestions provided by the reviewers. The manuscript has been revised to improve clarity, structure, and alignment with PLOS ONE’s submission guidelines. We believe the revisions have enhanced the manuscript and look forward to your feedback.

Sincerely,

Demba Keita, MD, DHSc, MPH

Epidemiological Supervisor

Sullivan County Department of Public Health

Demba.Keita@sullivanny.gov or demba.aja84@gmail.com

---

## [Decision Letter · Decision Letter 1]

17 Jul 2025

Quality of Life in Older Immigrant Adults on Hemodialysis

PONE-D-25-14166R1

Dear Dr. Keita,

We’re pleased to inform you that your manuscript has been judged scientifically suitable for publication and will be formally accepted for publication once it meets all outstanding technical requirements.

Kind regards,

Jordan Llego, PhD ELM, D. Hon. Ex., PhDN, RN

Academic Editor

PLOS ONE

Additional Editor Comments (optional):

Thank you for submitting your revised manuscript to PLOS ONE. We appreciate the time and effort you dedicated to addressing the reviewers' comments and strengthening the overall quality of your manuscript.

After careful review of your revised submission and the comprehensive responses to reviewer feedback, we are pleased to inform you that your manuscript has been accepted for publication in PLOS ONE.

We commend you on conducting a study of significant cultural and clinical relevance. Your work makes a valuable contribution to the growing body of literature that addresses health disparities and promotes holistic, patient-centered care for immigrant populations living with chronic illness.

Our production team will now proceed with preparing your manuscript for publication. You may be contacted for final proofs and typesetting.

Reviewers' comments:

Reviewer's Responses to Questions

**Comments to the Author**

Reviewer #1: All comments have been addressed

2. Is the manuscript technically sound, and do the data support the conclusions?

Reviewer #1: Yes

3. Has the statistical analysis been performed appropriately and rigorously?

Reviewer #1: N/A

4. Have the authors made all data underlying the findings in their manuscript fully available?

Reviewer #1: Yes

5. Is the manuscript presented in an intelligible fashion and written in standard English?

Reviewer #1: Yes

Reviewer #1: Thank you for your careful and thoughtful revision of the manuscript. The revised version shows significant improvements across multiple dimensions:

Sample Size Justification: You have adequately explained your rationale using the concept of “information power,” which is appropriate for qualitative inquiry. This strengthens the methodological transparency of your work.

Data Availability: Your update to the Data Availability Statement—indicating that data will be deposited in a public repository, subject to ethical restrictions—is consistent with PLOS ONE’s data policy and commendable in addressing participant privacy.

Thematic Depth and Cultural Context: The expanded discussion on cultural health practices (e.g., Ramadan fasting) and illness perceptions now offers greater contextual insight into the lived experiences of older immigrant adults on hemodialysis. This enriches the cultural and clinical relevance of your findings.

Use of Participant Quotations: The inclusion of detailed and representative participant narratives adds credibility and emotional resonance to the themes, elevating the quality of the qualitative analysis.

Clinical Implications: The revised discussion provides practical, actionable recommendations for culturally competent care, enhancing the impact of the research for healthcare providers and policymakers.

Formatting and Clarity: Structural improvements, standard English, appropriate referencing, and adherence to submission guidelines have further improved the readability and professionalism of the manuscript.

**Do you want your identity to be public for this peer review?** For information about this choice, including consent withdrawal, please see our Privacy Policy

Reviewer #1: No

---

## [Editor Report · Acceptance letter]

PONE-D-25-14166R1

PLOS ONE

Dear Dr. Keita,

I'm pleased to inform you that your manuscript has been deemed suitable for publication in PLOS ONE. Congratulations! Your manuscript is now being handed over to our production team.

Kind regards,

on behalf of

Dr. Jordan Llego

Academic Editor

PLOS ONE